
# 1    Aerosol absorption in global models from AeroCom Phase III

Maria Sand[1], Bjørn H. Samset[1], Gunnar Myhre[1], Jonas Gliß[2], Susanne E. Bauer[3,4], Huisheng Bian[5,6], Mian
Chin[6], Ramiro Checa-Garcia[7], Paul Ginoux[8], Zak Kipling[9], Alf Kirkevåg[2], Harri Kokkola[10], Philippe Le
Sager[11], Marianne T. Lund[1], Hitoshi Matsui[12], Twan van Noije[11], Samuel Remy[13], Michael Schulz[2], Philip
Stier[14], Camilla W. Stjern[1], Toshihiko Takemura[15], Kostas Tsigaridis[4,3], Svetlana G. Tsyro[2], and Duncan
Watson-Parris[14]
[1]CICERO Center for International Climate Research, Oslo, Norway
[2]Norwegian Meteorological Institute, Oslo, Norway
[3]NASA Goddard Institute for Space Studies, New York, USA
[4]Center for Climate Systems Research, Columbia University, New York, USA
[5]Maryland Univ. Baltimore County (UMBC), Baltimore, MD, USA
[6]NASA Goddard Space Flight Center, Greenbelt, Maryland, USA
[7]Laboratoire des Sciences du Climat et de l'Environnement, LSCE/IPSL, CEA-CNRS-UVSQ, Gif sur Yvette Cedex, France
[8]NOAA, Geophysical Fluid Dynamics Laboratory, Princeton, NJ, USA
[9]European Centre for Medium-Range Weather Forecasts, Reading, UK
[10]Atmospheric Research Centre of Eastern Finland, Finnish Meteorological Institute, Kuopio, Finland
[11]Royal Netherlands Meteorological Institute, De Bilt, the Netherlands
[12]Graduate School of Environmental Studies, Nagoya University, Nagoya, Japan
[13]HYGEOS, Lille, France
[14]Atmospheric, Oceanic and Planetary Physics, Department of Physics, University of Oxford, Oxford, UK
[15]Research Institute for Applied Mechanics, Kyushu University, 6-1 Kasuga-koen, Kasuga, Fukuoka, Japan
*Correspondence to*: Maria Sand (maria.sand@cicero.oslo.no)
**Abstract.** Aerosol induced absorption of shortwave radiation can modify the climate through local atmospheric heating, which
affects lapse rates, precipitation, and cloud formation. Presently, the total amount of such absorption is poorly constrained, and
the main absorbing aerosol species (black carbon (BC), organic aerosols (OA) and mineral dust are diversely quantified in
global climate models. As part of the third phase of the AeroCom model intercomparison initiative (AeroCom Phase III) we
here document the distribution and magnitude of aerosol absorption in current global aerosols models and quantify the sources
of intermodel spread. 15 models have provided total present-day absorption at 550 nm, and 11 of these models have provided
absorption per absorbing species. The multi-model global annual mean total absorption aerosol optical depth (AAOD) is
0.0056 [0.0020 to 0.0097] (550 nm) with range given as the minimum and maximum model values. This is 31% higher
compared to 0.0042 [0.0021 to 0.0076] in AeroCom Phase II, but the difference/increase is within one standard deviation
which in this study is 0.0024 (0.0019 in Phase II). The models show considerable diversity in absorption. Of the summed
component AAOD, 57 % (range 34-84%) is estimated to be due to BC, 30 % (12-49%) is due to dust and 14% (4-49%) is due
to OA, however the components are not entirely independent. Models with the lowest BC absorption tend to have the highest
OA absorption, which illustrates the complexities in separating the species. The geographical distribution of AAOD between



the models varies greatly and reflects the spread in global mean AAOD and in the relative contributions from individual
species. The optical properties of BC are recognized as a large source of uncertainty. The model mean BC mass absorption
coefficient (MAC$_{BC}$) value is 9.8 [3.1 to 16.6] m$^2$ g$^{-1}$ (550 nm). Observed MAC values from various locations range between
5.7-20.0 m$^2$ g$^{-1}$ (550 nm). Compared to retrievals of AAOD and absorption Ångstrøm exponent (AAE) from ground-based
observations from the Aerosol Robotic Network (AERONET) stations, most models underestimate total AAOD and AAE.
The difference in spectral dependency between the models is striking.

## 1 Introduction

Aerosols directly affect the energy budget of the atmosphere by interacting with solar radiation. While all aerosols scatter
shortwave radiation, some also absorb it, which in turn modifies the thermal structure of the surrounding air masses
(McCormick and Ludwig, 1967). This localized atmospheric heating can lead to rapid changes in dynamics, clouds and
precipitation (Hansen et al., 1997; Ackerman et al., 2000). The concentrations of (absorbing) aerosols vary greatly temporally
and spatially, due to their diverse and intermittent emission sources (e.g., forest fires) and short atmospheric lifetimes (days to
1-2 weeks). The ability of an aerosol to absorb solar radiation depends on its composition, mixing state, component refractive
indices, size and shape, which can also change during its lifetime. The dominant absorbing aerosol is black carbon (BC),
followed by mineral dust and organic carbon-based aerosols (OA) or brown carbon (BrC). The three absorbing species are
rarely observed as single species, while many models are not able to fully mix the aerosols and therefore treat them as separate
species in an idealized way with their own life cycles and optical properties.
BC, emitted from incomplete combustion processes, is a particularly strong absorber of solar radiation and absorbs across the
entire solar spectrum (Bond et al., 2013). BC quickly mixes with other aerosols and often becomes coated. This process
enhances the effective absorptivity of BC over time and is often referred to as 'aging' (Cappa et al., 2012). Some climate
models use a constant enhancement factor of 1.5 to define absorption of aged BC relative to freshly emitted BC (Bond and
Bergstrom, 2006). Models that include internal mixing of aerosols can calculate the absorption enhancement based on the
mixing state, but these calculations are approximate (using mixing rules or the assumptions of a co-centric core/shell structure)
(Stier et al. 2017).
However, these calculations rely on reliable representations of the aerosol mixing state as well as on assumptions in the
calculation of the radiative properties itself, such as effective medium approximations or core/shell models (c.f. Stier et al,

63  2007).



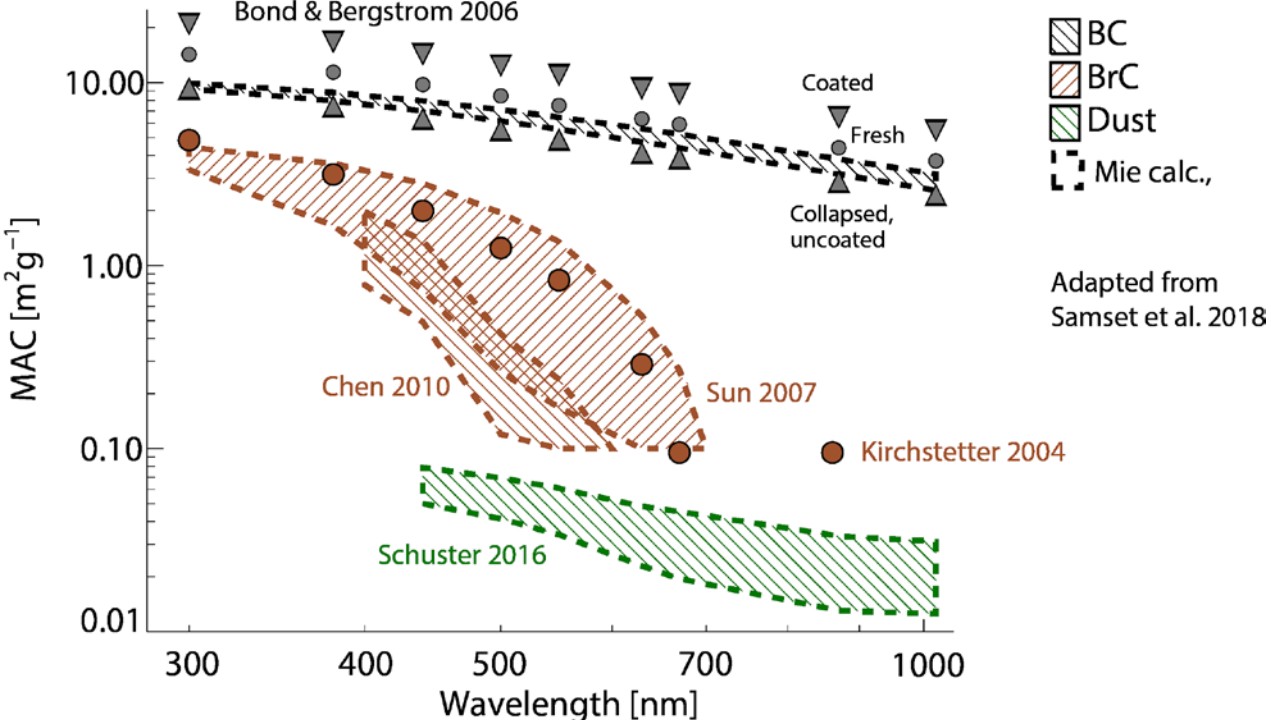

**Figure 1: Per-species mass absorption coefficient (MAC) as function of wavelength, from observations and radiative transfer calculations. BC, BrC/OA and dust can be seen to have separable properties, which underlies the usage of these species as emitted, transported and radiatively active particle types in most global climate models. Adapted from Samset et al (2018).**

Mineral dust is one of the most abundant aerosols by mass and in AeroCom Phase I and III close to 60-70% of the dry mass (Textor et al., 2006, Gliß et al., 2021), but has a much lower imaginary part of the refractive index compared to BC and absorbs less per mass (Sokolik and Toon, 1999). Absorption also depends on particle size distribution. While fine-dust particles mostly scatter solar radiation, coarse dust also absorbs moderately in the visible and near-infra-red spectrum. Models tend to substantially underestimate (or even neglect) the amount of coarse dust particles (with diameter $\geq$ 5 μm) in the atmosphere and very large particles are rarely represented in models (Adebiyi and Kok, 2020; Kok et al., 2017). This bias may imply that models underestimate the absorption by mineral dust, at least in the long-wave spectrum (Lacagnina et al., 2015). However, the constraints in the current dust emissions schemes makes the models reproduce dust optical depth reasonably well (Ridley et al. 2016), with a consistent regional seasonal cycle when compared with satellite observations, and AERONET local measurements well compared over dusty stations (Pu and Ginoux, 2018; Checa-Garcia et al, 2020). Absorption also depends on dust mineralogical composition, in which different minerals absorb stronger or weaker and have a distinct wavelength dependence, something that is missing in most climate models (e.g., Perlwitz et al., 2015). Iron oxides (hematite and goethite) are minerals that enhance the absorption. The presence of these minerals depends on the parent soil, and specific deserts have different fractions of minerals.



Carbonaceous aerosols can also include weakly absorbing organic compounds (Andreae and Gelencsér, 2006). The
absorptivity of organic aerosols decreases rapidly with increasing wavelength in the solar to UV spectrum (Kirchstetter et al.,
2004), as shown in Fig. 1. BC is often coated with these organic aerosols and a thorough conceptual separation between the
two aerosol types is difficult (Jacobson et al., 2000). Since the total aerosol absorption depends on the composition, size and
shape of aerosols, all of which vary greatly, the magnitude of aerosol absorption is highly uncertain, both from a measurement
perspective and in general circulation models (Haywood and Shine, 1995; Cooke and Wilson, 1996; Moosmüller et al., 2009).

The multi-model initiative 'Aerosol Comparisons between Observations and Models' (AeroCom) assesses state-of-the-art
aerosol modelling to better understand global aerosols and their impact on climate (https://aerocom.met.no) (Schulz et al.,
2006; Kinne et al., 2006; Textor et al., 2007; Koch et al., 2009). The models use a common protocol and are encouraged to use
identical emission inventories for prescribed emissions. In the previous AeroCom phase II experiment, the total direct radiative
forcing was estimated at $-0.27$ Wm$^{-2}$ from 16 models. (Myhre et al., 2013). The present-day absorption aerosol optical depth
(AAOD) at 550 nm was estimated at 0.0042, with a range of [0.0021, 0.0076] (Samset et al., 2018). Table S1 in supplement
provides numbers for the individual models used in AeroCom Phase II. In this study we use the term absorption to describe
absorption optical depth and not atmospheric absorption, which is a convergence of radiative fluxes between the TOA and the
surface (in Wm$^{-2}$). The latter depends on clouds and surface albedo in the models (Stier et al 2013; Randells et al. 2013).
Gliß et al. (2021) compared modelled optical properties in AeroCom Phase III with a wide range of remote sensing and in-situ
observations. They found that most models underestimate total column AOD as well as "dry" (i.e., below RH<40%) surface
scattering and absorption coefficients, suggesting that aerosol loadings might be underestimated. A comparison with
AERONET measurements of the Ångström Exponent (AE) suggested that models overestimate size or underestimate the fine
mode fraction, but the separation into fine (< 1 um) and coarse mode (> 1 um) AOD indicated that the same behaviour does
not apply for this specific size-segregation.
To further investigate these issues, we here present aerosol absorption in 15 state-of-the-art aerosol models from AeroCom
Phase III. We aim to better quantify the sources of model spread by separating absorption per species (BC, OA, and dust) and
investigate regional and seasonal differences.
**2 Methods**
**2.1 AeroCom models**
Table 1 and 2 summarises the models used in this paper. The models have provided monthly mean values for 2010 and 1850
using the same prescribed anthropogenic and biomass burning emission datasets when possible and with fixed sea surface
temperatures. Some models also applied atmospheric nudging to 2010 meteorology. Anthropogenic fossil fuel, biofuel and





biomass burning emissions are from the Community Emission Data System (CEDS) (Hoesly et al., 2018) and from the
historical global biomass burning emissions for CMIP6 (van Marle et al., 2017). It is only BC emissions among the absorbing
species that are consistent among the models. The global model-mean 2010 BC emissions amount to 9.6 Tg/yr (model range
9.1 to 9.8 Tg/yr), while dust emissions, which are calculated online in most models based on modeled climate, range from 848
- 5646 Tg/yr with a model-mean of 1771 Tg/yr and OA emissions vary from 48 - 158 Tg/yr with a model-mean of 91.4 Tg/yr.
The differences in primary OA emissions are caused by different OA/OC ratios (see Table 2) and the fact that some models
include marine emissions and a few models also include SOA emissions (even though SOA is not primary emissions).  15
models have provided total absorption at 550 nm and 11 models have provided absorption split into BC and dust (OA). As
shown in Table 2 there are differences in mixing assumptions. A few models assume fully externally mixed aerosols, while
most models assume partly internal mixing, using different mixing rules for calculating the refractive indices. An overview of
the refractive indices separated into the real and imaginary parts for BC, OA, and dust in the AeroCom models are shown in
Fig. 2. The real part of the refractive index indicates scattering (and increased scattering for high values) while high values of
the imaginary index indicate (high) absorption. OsloCTM3 divide OA into a mix of absorbing and non-absorbing species
(which is why the imaginary part of the refractive index is relatively large). Most models have reported clear-sky AAOD,
while some models have assumed all-sky conditions (EMEP, GEOS, GFDL, and OsloCTM3). The AeroCom model mean was
computed by regridding the models on a 2°×3° resolution before averaging. A comprehensive description of the AeroCom
Phase III models is given in Gliß et al. (2021). Note that the same "AeroCom control" model experiment was used in the
present study and Gliß et al. (2021) and that the aerosol life cycle properties (emissions, lifetime, burden) and optical properties
are consistent in between the two studies.
**Table 1: AeroCom Phase III model description**

| Model | Label for model and simulation setup | Resolution | References |
|---|---|---|---|
| CAM5-ATRAS | CAM5-ATRAS_AP3-CTRL | 1.9 × 2.5, 30 levs | Matsui (2017); Matsui and Mahowald, (2017) |
| EC-Earth3 | EC-Earth3-AerChem-met2010_AP3-CTRL2019 | 2.0 × 3.0, 34 levs | van Noije et al. (2014); van Noije et al. (submitted) |
| ECHAM-HAM | ECHAM6.3-HAM2.3-met2010_AP3-CTRL | 1.9 × 1.9, 47 levs | Tegen et al. (2019) |
| ECHAM-SALSA | ECHAM6.3-SALSA2.0-met2010_AP3-CTRL | 1.9 × 1.9, 47 levs | Kokkola et al. (2018) |
| ECMWF-IFS | ECMWF-IFS-CY46R1-CAMS-CTRL-met2010_AP3-CTRL | 0.4 × 0.4 | Rémy et al. (2019) |
| EMEP | EMEP_rv4_33_Glob-CTRL | 0.5 × 0.5, 20 levs | Simpson et al. (2012) |
| GEOS | GEOS-i33p2-met2010_AP3-CTRL | 1.0 × 1.0, 72 levs | Colarco et al. (2010) |





| GFDL | GFDL-AM4-met2010_AP3-CTRL | 1.0 × 1.2, 33 levs | Zhao et al. (2018) |
|------|---------------------------|---------------------|----------------------|
| GISS-OMA | GISS-ModelE2p1p1-OMA_AP3-CTRL | 2.0 × 2.5, 40 levs | (Bauer et al, 2020; Koch, 2001) |
| GISS-MATRIX | GISS-ModelE2p1p1-MATRIX_AP3-CTRL | 2.0 × 2.5, 40 levs | (Bauer et al, 2008) |
| INCA | INCA_AP3-CTRL | 1.3 × 2.5, 79 levs | (Balkanski et al., 2004; Schulz et al., 2009) |
| NorESM2 | NorESM2-met2010_AP3-CTRL | 0.9 × 1.2, 32 levs | Kirkevåg et al. (2018); Seland et al. (2020) |
| OsloCTM3 | OsloCTM3v1.02-met2010_AP3-CTRL | 2.25 × 2.25, 60 levs | Myhre et al. (2007); Lund et al.. (2018) |
| SPRINTARS | MIROC-SPRINTARS_AP3-CTRL | 0.6 × 0.6, 56 levs | Takemura et al. (2005) |
| TM5 | TM5-met2010_AP3-CTRL2019 | 2.0 × 3.0, 34 levs | Bergman et al., (in preparation); van Noije et al. (submitted) |



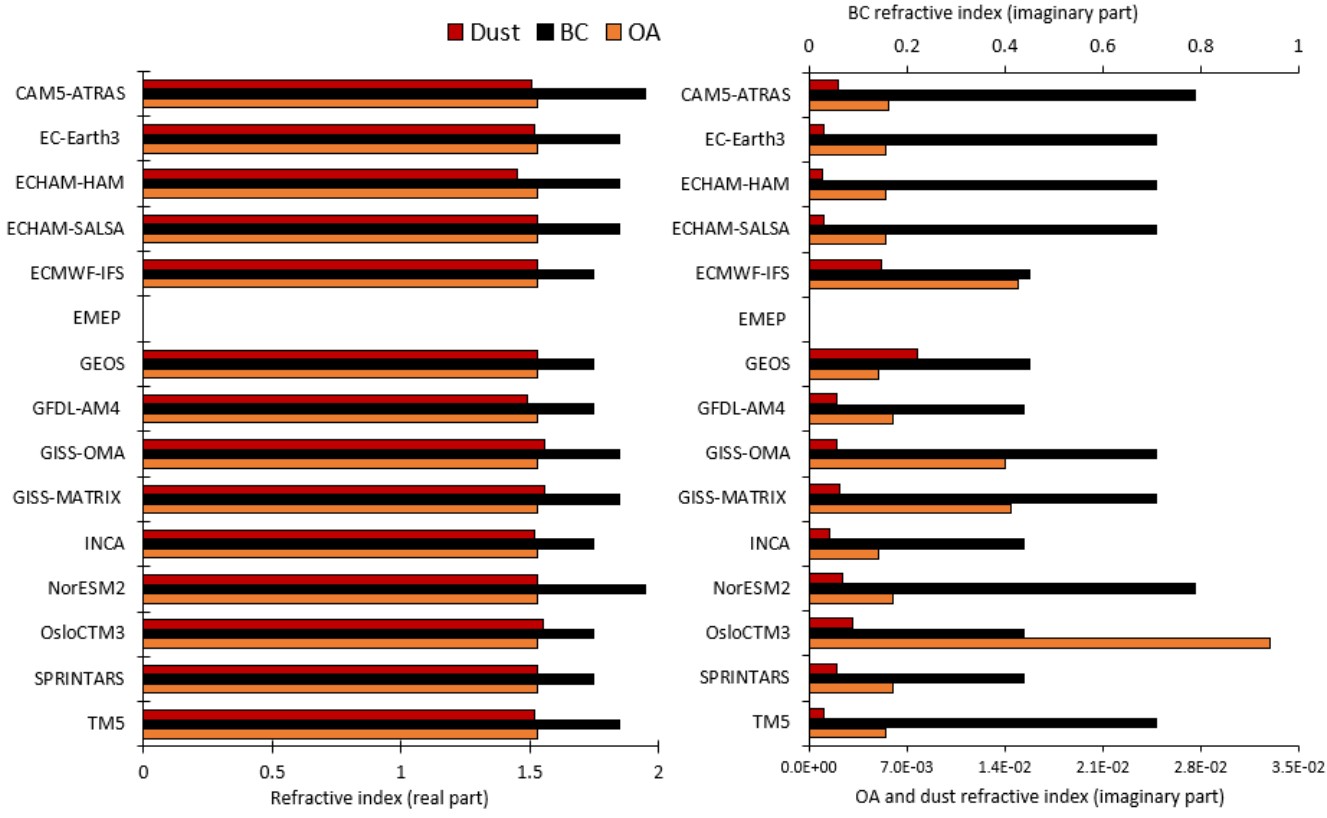




**Figure 2: Refractive index (real part (left) and imaginary part (right) for the AeroCom models for BC (black bars), OA (orange) and dust (red). Note the different axes on the right panel. EMEP has bulk mass and does not calculate refractive index. The numbers are also given in Table S2 in Supplement.**

**Table 2: Overview of the mixing assumptions in the models**

| Model | Mixing assumptions | Method for splitting absorption into individual contributions (if internally mixed): | OA/OC ratio |
|---|---|---|---|
| CAM5-ATRAS | Core-shell for internally-mixed BC particles; Volume mixing for pure BC and BC free particles. | Absorption per species is calculated from the difference of absorption between optical (Mie theory) calculations considering all aerosols species and all aerosol species except the target species. | 1.4 |
| EC-Earth3 | Sulfate, ammonium-nitrate, organic aerosols, sea salt, and water treated as homogeneous mixtures described by the Bruggeman mixing rule. Maxwell–Garnett mixing rule for BC and dust present in mixture. | - | 1.6 |
| ECHAM-HAM | Internal and external mixing of log-normal modes using volume weighting of refractive indices (alternative mixing rules Bruggeman and Maxwell-Garnett available but have limited impact). | Component absorption aerosol optical depth is approximated from total aerosol absorption optical depth through volume and imaginary part of the refractive index weighting of individual compounds. | 1.4 |
| ECHAM-SALSA | Internal and external mixing using volume weighting of refractive indices. | The aerosol absorption optical depth is weighted by volume and the imaginary part of the refractive index of individual compounds. | 1.4 |
| ECMWF-IFS | External mixing | - | 1.8 |
| EMEP | External mixing | - | 1.25 FF, 1.67 BB |
| GEOS | External mixing | - | 1.8 |
| GFDL | All aerosols externally mixed, except for SO4 and BC which are internally mixed by volume weighting of refractive indices, including hygroscopic growth of SO4 | The volume of BC, SO4 and ambient RH in each grid cell every 3 hours is used to extract the closest values of Qext, SSA, ASYM from a look-up table to calculate the radiative fluxes | 1.4 |
| GISS-OMA | External mixing. Dust coating with sulfate and nitrate only affects dust lifetime. BC absorption amplification of 1.5. | - | 1.4 |
| GISS-MATRIX | Internal mixing, by tracking populations defined by mixing state | - | 1.4 |



| INCA | External mixing except BCin soluble mode which is internally mixing with SO4. Maxwell-Garnett mixing rule to compute its refractive index (Wang, R et al 2016). | In the mixing rule the volume fraction of BC inclusions and the refractive index of the non-absorbing soluble specie change according to the simulated composition of the soluble accumulation mode and atmospheric relative humidity. | 1.4 |
|---|---|---|---|
| NorESM2 | Internal and external mixing. Maxwell-Garnett is used for calculation of refractive index of internal mixing of BC with other components, otherwise volume mixing. | The fraction of the aerosol extinction (scattering and/or absorption) for a given species and size-bin is reported by computing the volume fraction of aerosol species in aerosol particle volume (without water) in that particular size-bin using the following densities (dust = 2650 kg/m3, sea salt = 1600 / SO4 = 1769 / BC = 1500 / POM = 1500) | 1.4 for FF, 2.6 for BB. |
| OsloCTM3 | BC internal mixing with non-scattering species. Internal mixing of BC and OA from biomass burning. External mixture for other aerosols. | All absorption is linked to BC | 1.8 for SOA; 1.6-1.8 for FF; 2.6 for BB. |
| SPRINTARS | External mixing, except 50% of BC from fuel sources is internally mixed with OC. The volume weighting of refractive indices is assumed for the internal mixture. | BC AAOD is calculated assuming all BC is externally mixed | 1.6 F; 2.6 BB. |
| TM5 | Sulfate, ammonium-nitrate, organic aerosols, sea salt, and water treated as homogeneous mixtures described by the Bruggeman mixing rule. Maxwell–Garnett mixing rule for BC and dust present in mixture. | - | 1.6 |


## 2.2 Observational data

We have compared modelled BC MAC with available observations found in literature (see Supplement for a complete list).
We define MAC in the models as the global mean BC AAOD at $\lambda = 550$ nm divided by the global mean column load of BC.
All observations have been converted to $\lambda = 550$ nm by assuming that the absorption Ångstrøm exponent (AAE) equals 1.
Total AAOD and AAE is compared to retrieved data from ground-based stations in the Aerosol Robotic Network (AERONET)
version 2.0 (https://aeronet.gsfc.nasa.gov/) (Holben et al. 1998; 2006; Dubovik et al., 2000). We have selected the AERONET
stations that have at least 25% daily coverage (i.e., at least 7 days) to compute AERONET monthly means from daily values.





## 3 Results

In this section we first present model results of the total aerosol absorption optical depth (AAOD) at 550 nm and the AAOD contributions from BC, OA and dust, followed by a comparison of the mass absorption coefficient (MAC) for BC to observed values, a discussion about the absorption Ångström exponent, and a comparison with AERONET AAOD.

### 3.1 Total AAOD in AeroCom Phase III

Figure 3 shows the total AAOD at 550 nm for the 15 AeroCom Phase III models. The global mean values range from 0.0020 (SPRINTARS) to 0.0097 (GISS-MATRIX) (Fig. 3a). The two models differ substantially in their treatment of aerosol absorption. In SPRINTARS, the aerosols are externally mixed. In GISS-MATRIX all aerosols are internally mixed, and populations are tracked by mixing state. Also, their imaginary parts of the refractive index vary a lot (1.75 + 0.44i for SPRINTARS and 1.85 + 0.71i for GISS-MATRIX (Fig. 2). AAOD values for all the models are given in Table 3. The multi-model mean is 0.0056, with a standard deviation of 0.0024. The multi-model mean is 31% higher compared to the previous multi-model mean in AeroCom Phase II (using emissions for year 2000) (Samset et al., 2018). In AeroCom Phase II, the model mean (using 14 models) is 0.0042, with range 0.0021 to 0.0076 and standard deviation 0.0019. The model range in total AAOD in AeroCom Phase III (0.0077) is larger than in Phase II (0.0055), but the spread (here defined as range/mean) is similar (1.5 and 1.3). AAOD for the different models in AeroCom Phase II is given in Table S1 in Supplement.

The spread is particularly large at NH mid latitudes (Fig. 3b). The seasonal cycle has maximum values during August and September, which is linked to biomass burning (Fig. 3c). The annual mean geographical distribution (Fig. 3d) shows strong absorption over Central Africa, linked to biomass burning, and a maximum in China and India linked to anthropogenic emissions. Geographical distributions of AAOD for all seasons are shown in the Supplement. In July, August and September, the onset of the biomass burning season in South America and Southern Africa is apparent, along with dust plumes from the Saharan desert. A weaker maximum is seen in several of the models in February and March linked to biomass burning in central Africa.





**Figure 3: Total AAOD at λ = 550 nm from the models; (a) annual global mean, (b) annual zonal mean (c) the global seasonal cycle and (d) annual mean spatial distributions.**

**Table 3. Total, BC, OA and dust AAOD at 550 nm, BC MAC (550 nm), BC burden, and BC, OA and dust lifetime**



| | Total AAOD | BC AAOD | OA AAOD | Dust AAOD | BC MAC | BC Burden | BC lifetime | OA lifetime | Dust lifetime |
|---|---|---|---|---|---|---|---|---|---|
| CAM5-ATRAS | 0.0034 | 0.0021 | 0.00062 | 0.0009 | 9.1 | 0.23 | 4.5 | 6.1 | 3.0 |
| EC-Earth3 | 0.0067 | - | - | - | - | 0.45 | 8.7 | 9.3 | 3.9 |
| ECHAM-HAM | 0.0042 | 0.0035 | 0.00018 | 0.0006 | 10.2 | 0.34 | 6.4 | 6.0 | 6.0 |
| ECHAM-SALSA | 0.0091 | 0.0077 | 0.00037 | 0.0011 | 15.0 | 0.51 | 9.6 | 8.2 | 7.0 |
| ECMWF-IFS | 0.0055 | - | - | - | - | 0.20 | 3.9 | 4.3 | 1.4 |
| EMEP | 0.0025 | 0.0014 | - | 0.0011 | 10.4 | 0.13 | 2.2 | 4.3 | 3.2 |
| GEOS | 0.0040 | 0.0016 | 0.00041 | 0.0020 | 7.8 | 0.21 | 4.1 | 4.6 | 5.4 |
| GFDL | 0.0084 | 0.0051 | 0.00087 | 0.0022 | 16.6 | 0.31 | 5.9 | 4.1 | 3.5 |
| GISS-MATRIX | 0.0097 | - | - | - | - | 0.22 | 4.2 | 5.1 | 7.8 |
| GISS-OMA | 0.0081 | 0.0022 | 0.00071 | 0.0021 | 10.0 | 0.22 | 4.2 | 6.3 | 5.4 |
| INCA | 0.0042 | 0.0021 | 0.00022 | 0.0018 | 7.5 | 0.28 | 5.5 | 6.0 | 4.5 |
| NorESM2 | 0.0039 | 0.0011 | 0.00155 | 0.0006 | 5.2 | 0.33 | 6.4 | 6.2 | 1.9 |
| OsloCTM3 | 0.0055 | 0.0037 | 0.00020 | 0.0017 | 12.4 | 0.23 | 4.4 | 5.3 | 3.4 |
| SPRINTARS | 0.0020 | 0.0007 | 0.00030 | 0.0007 | 3.1 | 0.23 | 5.1 | 3.4 | 2.3 |
| TM5 | 0.0064 | - | - | - | - | 0.44 | 8.6 | 8.8 | 4.0 |
| | | | | | | | | | |
| Mean | 0.0056 | 0.0028 | 0.00054 | 0.0013 | 9.8 | 0.29 | 5.6 | 5.9 | 4.2 |
| Median | 0.0055 | 0.0021 | 0.00039 | 0.0011 | 10.0 | 0.23 | 5.1 | 6.0 | 3.9 |

179 BC MAC in [$m^2 g^{-1}$]; BC burden in [$mgm^{-2}$], lifetime in days. The mean and median in this table is calculated from the global mean of each model (in this table), and is not the same
180 as the AEROCOM-MEDIAN field shown in Fig 3, 6,7 and 8.

181

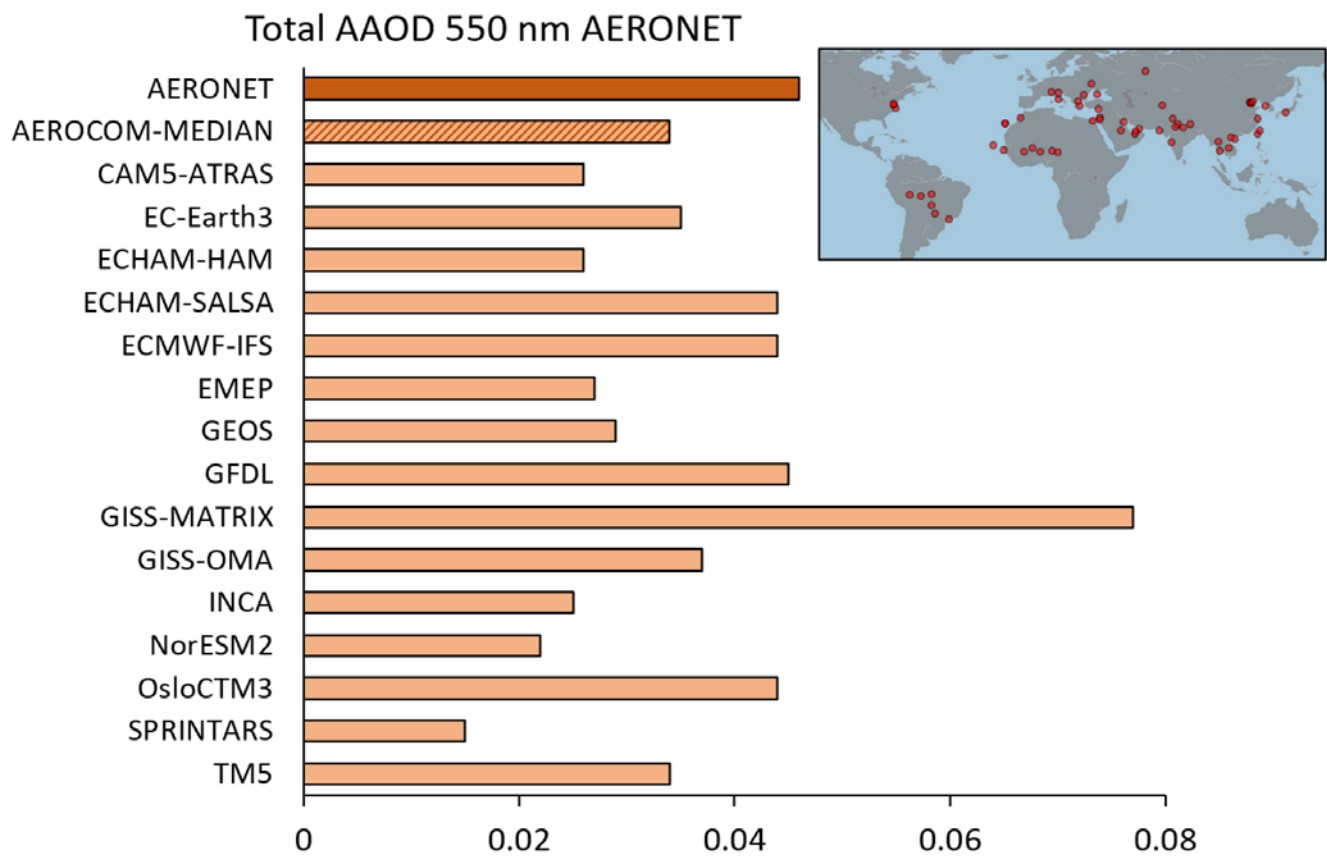

**Figure 4: AeroCom AAOD at λ = 550 nm at the nearest grid point compared to AERONET retrieved AAOD averaged over stations requiring 25% daily coverage to compute AERONET monthly means from daily values (shown in the map).**

Figure 4 shows a comparison between the AeroCom total AAOD at the nearest grid point to retrieved AERONET AAOD from sun photometers at the stations that have at least 25% daily coverage to compute AERONET monthly means from daily values (i.e., at least 7 days, stations shown in the map) for year 2010. The AERONET mean AAOD is 0.046. All models except GISS-MATRIX show lower values, even though a few models are close to this value. The AeroCom model average is 0.035 (range 0.015-0.077) at the selected AERONET sites. Months with no observations are excluded prior to averaging. Seasonal cycle at 6 stations influenced by dust (Canary Islands), biomass burning (South America) and industrial emissions (China close to Beijng) is shown in Supplement Fig. S2.

Some caution needs to be exercised when comparing AAOD from models with AERONET. To minimize the uncertainties in the retrieval of AAOD Level 2, AOD is required to be larger than 0.4 at 440 nm and the solar zenith angle must be larger than 50° (Dubovik et al. 2000). This means that AERONET AAOD is skewed towards high aerosol loadings. Since model data are available only at monthly resolution, a corresponding exclusion of days with low AOD could not be done here. Another limitation of this comparison is the coarse resolution of the model data (1°×1° resolution, but there are also models with lower





resolutions that were interpolated) compared to point measurements from AERONET with a narrow field of view. This
complicates a comparison since AERONET sites generally are located close to aerosol sources, and this may cause a global
representation error up to 30 % (Wang et al., 2018). However, using a high-resolution simulation of global aerosols, Schutgens
(2020) found a much smaller bias of 9 %.

**3.2 Absorption of BC, OA and dust**

The relative absorption varies between BC, OA and dust in the models. Figure 5 shows the distribution of total absorption
between the three species. The models with internally mixed aerosols have different methods for splitting the total absorption
into individual contributions (Table 2). For internal mixtures, this is conceptually difficult (and the approximations described
above have strong limitations). The component AAODs are therefore not independent. Absorption of BC accounts for on
average 57% of total absorption [with a range 34-84%]. The absorption of OA accounts for 14% [4-49%]. The models with
the smallest portion of BC absorption (NorESM2, SPRINTARS and GISS-OMA) have the highest portion of OA absorption.
GISS-OMA has one of the highest imaginary parts in the OA refractive index across all models, to implicitly account for some
browniness in OA (Tsigaridis and Kanakidou, 2018). Dust absorption accounts for 30% [12-49%].
The thin bar represents the total AAOD. For four models (CAM5-ATRAS, GFDL, NorESM2 and SPRINTARS), the total
AAOD deviates from the sum of BC, OA, and dust AAOD. In CAM5-ATRAS, the reason for the deviation is that AAOD per
species is calculated from the difference of absorption between optical calculations considering all aerosols species and all
aerosol species except the target species. In NorESM2 the additional absorption is from sea-salt and sulfate (mixed with BC,
dust and OA). In GFDL BC is internally mixed with $SO_4$, so the additional absorption is due to $SO_4$ (mixed with BC), including
hygroscopic growth. This part is marked with grey color in the BC absorption bar. In SPRINTARS, the individual contribution
is calculated assuming external mixture.





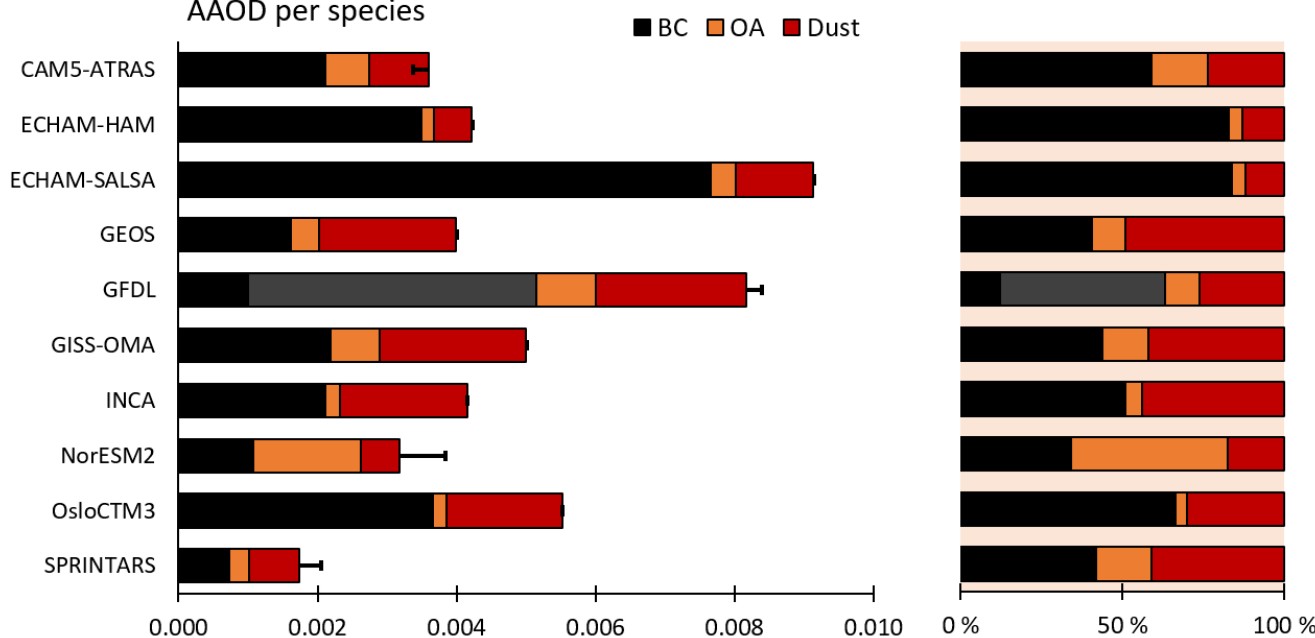

**Figure 5: Global mean AAOD at λ = 550 nm for BC (black), OA (orange) and dust (red); absolute values on the left and relative values on the right. The thin bar shows the total AAOD. The grey area in GFDL is BC mixed with SO₄.**

Figure 6 shows the AAOD for BC at 550 nm for 11 models. The multi-model global mean is 0.0028. Here, the AeroCom models show a large range in values from 0.0007 (SPRINTARS) to 0.0077 (ECHAM-SALSA). ECHAM-SALSA has the highest BC burden (0.51 mg m$^{-2}$, see Table 3) and longest lifetime (9.6 days) among the models, while the BC burden in SPRINTARS is in the lower range (0.3 mg m$^{-2}$). For ECHAM-SALSA the BC burden and lifetime has shown to be very sensitive to wet deposition and assumptions on the mixing of BC with other compounds (Holopainen et al., 2020). The models with the longest lifetime of BC also place more BC aloft compared to the other models (Fig. S4). The spread in BC burden is lower between the models compared to the spread in BC AAOD (relative standard deviation for BC burden is 0.38 compared to 0.66 for BC AAOD) (Fig. S4). The models that assume external mixing (EMEP, GEOS, GISS-OMA and SPRINTARS) generally show the lowest BC absorption, except NorESM2, however this model report part of the BC absorption as SO₄ and sea-salt absorption since these two species are partly internally mixed with OA and dust, (following old AeroCom protocol recommendations, https://aerocom.met.no/protocol_expl.html), and this is not reported in Fig. 6. For GFDL we have here shown the value for BC only, and not BC mixed with SO4 (grey bar in Fig 5), and in Table 3 the total value is shown (0.0051). Most models show maximum absorption during early autumn (Fig. 6 (c). This is linked to the biomass burning season in Southern Africa and South America. The anthropogenic signal in China and India is apparent all year round.

**Figure 6: BC AAOD at λ = 550 nm from the models; (a) annual global mean, (b) annual zonal mean (c) the global seasonal cycle and (d) annual mean spatial distributions.**

Figure 7 shows the absorption of OA at 550 nm for 10 models. The global model-mean is 0.00054 with a range [0.00018 to 0.00155]. The global model-median is considerably lower than the mean; 0.00039 (Table 3). NorESM2 has a much larger absorption of OA compared to the other models. This is also the model with the second smallest absorption of BC. This is due to internal mixing of BC and OA in the model, where NorESM2 typically places more weight on OA relative to other models.





This illustrates the complexities of dividing between OA and BC (and dust) in models where aerosols are internally mixed.
The maximum values of OA absorption are linked to the biomass burning season in the southern hemisphere in late summer
and autumn. Part of the spread of OA absorption can be linked to a high diversity in OA emissions (48 - 246 Tg) since the
models have different parameterizations applied to ratio of OA to organic carbon (OC) and to secondary organic aerosol
formation and marine emissions, in addition to different refractive indices and mixing assumptions.



**Figure 7: OA AAOD at λ = 550 nm from the models; (a) annual global mean, (b) annual zonal mean (c) the global seasonal cycle and (d) annual mean spatial distributions.**

Figure 8 shows the absorption of mineral dust for 11 models. The global model-mean dust AAOD is 0.0013 (550 nm) which is approximately half of the BC AAOD. The values range from 0.0006 to 0.0022. Dust emissions in the models are a function





of wind speed and soil wetness/humidity and vegetation type. Current models do not implement explicit mineralogy, otherwise
optical properties would depend on soil properties with different mineral fractions. The models show a maximum in dust
absorption over the largest sources from Sahara and deserts in East Asia, peaking during spring and summer. The three models
with the lowest dust AAOD (ECHAM-HAM, SPRINTARS and NorESM2) show much lower dust absorption over the Sahara
Desert and Atlantic outflow region during spring (Fig. S6). SPRINTARS and NorESM2 have the lowest dust mass column
burden compared to the other models, while this is not the case for ECHAM-HAM (Fig. S8). The low dust loadings for
NorESM and SPRINTARS are due to both the short lifetime of dust (1.9 and 2.3 days compared to model mean 4.3 days) and
lower dust emissions compared to the other models.



**Figure 8: Dust AAOD at λ = 550 nm from the models; (a) annual global mean, (b) annual zonal mean (c) the global seasonal cycle and (d) annual mean spatial distributions.**



### 3.3 BC MAC values

Figure 9 shows the global mean $MAC_{BC}$ values in the AeroCom models. We define MAC here as the global mean BC AAOD divided by the global mean column load of BC. The $MAC_{BC}$ values range from 3.1 $m^2$ $g^{-1}$ (SPRINTARS) to 16.6 $m^2$ $g^{-1}$ (GFDL). Due to varying amounts of non-absorbing components (e.g., $SO_4$) attached to BC particles, it is difficult to report a clearly defined value of $MAC_{BC}$ for models with internal mixing. GFDL and NorESM2 have two reported $MAC_{BC}$ values. The lighter coloured bar for GFDL represents absorption of BC mixed with $SO_4$. This is in line with how the other models with internal mixing report absorption for BC (e.g., OsloCTM3). For NorESM2 the lighter coloured bar represents absorption of BC+OA+dust mixed with $SO_4$ and sea salt (which in the model does not mix internally with dust). The model-mean $MAC_{BC}$ value is 9.8 $m^2$ $g^{-1}$. (and 8.4 $m^2$ $g^{-1}$ if the conservative estimates for NorESM2 and GFDL are used).

An earlier proposed $MAC_{BC}$ value is 7.5 $m^2$ $g^{-1}$ (550 nm) for freshly generated BC and up to 11 $m^2$ $g^{-1}$ for aged BC (Bond and Bergstrom, 2006). Zanatta et al. (2016) suggested near-surface values for Europe between 9.1 to 20 $m^2$ $g^{-1}$ (converted to 550 nm). Lower $MAC_{BC}$ values (550 nm), 5.7, are found in the Arctic (Ytrri et al. 2014). The black dots in Fig 8 shows all available observations/estimates of $MAC_{BC}$ converted to $\lambda = 550$ nm (see Methods). The average of all selected values in this study is 10.9 $m^2$ $g^{-1}$ and a standard deviation of 3.1 $m^2$ $g^{-1}$. Please note that the models show column integrated global mean values, and they are not co-located with the locations of the observed $MAC_{BC}$ values. Assuming that the model values and observed values are still comparable (which is not obvious), SPRINTARS and NorESM2 are located outside the observed $MAC_{BC}$ range.

MAC values for OA and dust are much lower than for BC (0.15 and 0.04 model mean respectively, see Fig. S8 in Supplementary).





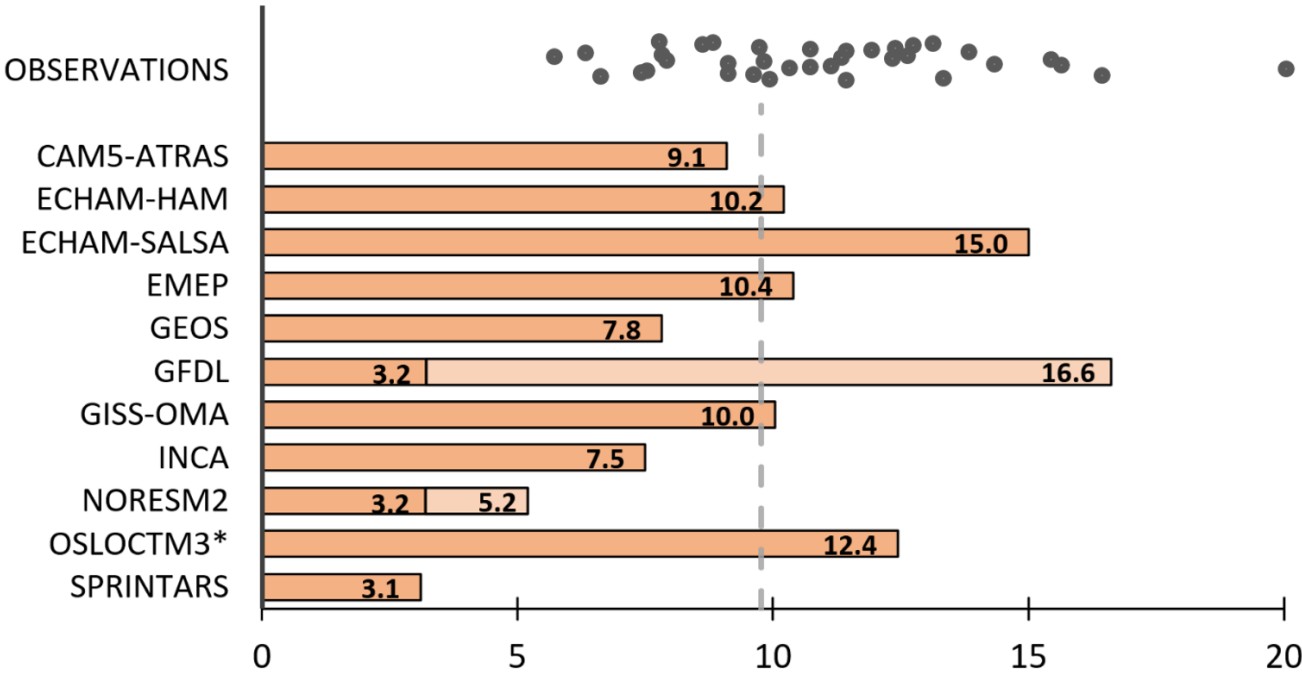

**Figure 9: Global mean MAC$_{BC}$ values at λ=550 nm for each model as atmospheric column integrated values. *OsloCTM3 is for BC from fossil fuels and biofuels only. The vertical striped line is the model mean. GFDL and NorESM2 have two reported MAC values (explanation in text). Black dots represent available observations in literature for various locations (converted to 550 nm). A list of the near-surface observations with references is found in the Supplement (Table S2).**

Figure 10 shows the variability in emissions, lifetime, and MAC with respect to AAOD of BC, OA, and dust for the AeroCom models. These 'partial sensitivities' are calculated by dividing the variable (emissions/lifetime/MAC) in each model by the AeroCom mean multiplied with the AAOD AeroCom mean. For BC, the variability in AAOD (*p_AAOD*) cannot be explained by different emissions, but in differences in MAC (which depends on both the aerosol microphysics scheme and on the method for estimating/approximating component specific AAODs) and lifetime, where especially two models (ECHAM-SALSA and EMEP) differ from the rest. For OA, some of the variability in AAOD can be explained by different emissions, lifetime, and MAC. For dust, the differences in lifetime and MAC are slightly higher than the variability in AAOD, suggesting there are compensating effects.



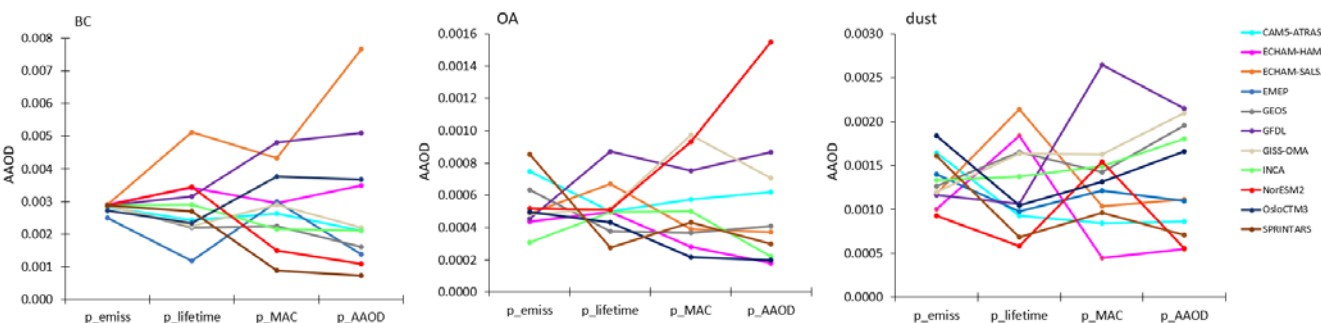

296

**Figure 10: Partial sensitivity of AAOD to variation in emission, lifetime, and MAC for BC, OA and dust for each model. The sensitivities are calculated by dividing the variable in each model by the AeroCom mean multiplied with the AAOD AeroCom mean.**

**3.4 Absorption at λ=440 nm and λ=870 nm**

Eight models (CAM5-ATRAS, ECHAM-HAM, GEOS, GFDL, INCA, SPRINTARS, NorESM2 and OsloCTM3) have also reported total absorption at λ=440 nm and ten models (the above plus GISS-MATRIX and GISS-OMA) have reported total absorption at λ=870 nm. The global, zonal, and seasonal mean is shown in Fig. S7 in Supplement. The model mean AAOD at 440 nm is 0.0060 [0.0025 – 0.0115]. The model mean AAOD at 870 nm is 0.0028 [0.0014 – 0.0047].

Figure 11 shows the contribution from BC, OA and dust to aerosol absorption at λ = 440 nm, 550 nm and 870 nm for the five models providing results per species at these wavelengths. The absorption is higher for 440 nm compared to 870 nm for all the species, which is in accordance with observations (Dubovik et al., 2002), even though the spectral dependence OA is notably low. The relative contribution from dust is higher for 440 nm compared to 870 nm. The relative contribution from OA is slightly larger for 870 nm, while for BC is it slightly lower for 440 nm compared to 870 nm.





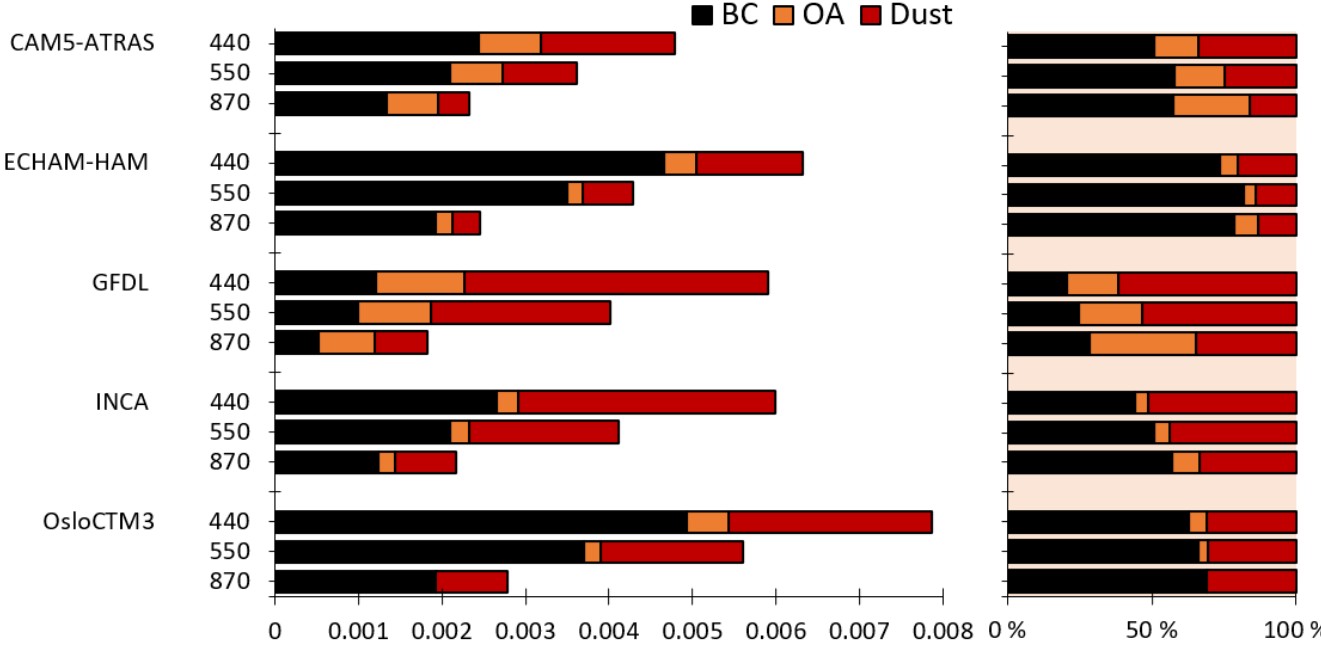

Figure 11: Global mean AAOD at λ = 440, 550 and 870 nm for each model split into BC (black), OA (orange) and dust (red); absolute values on the left and relative values on the right.



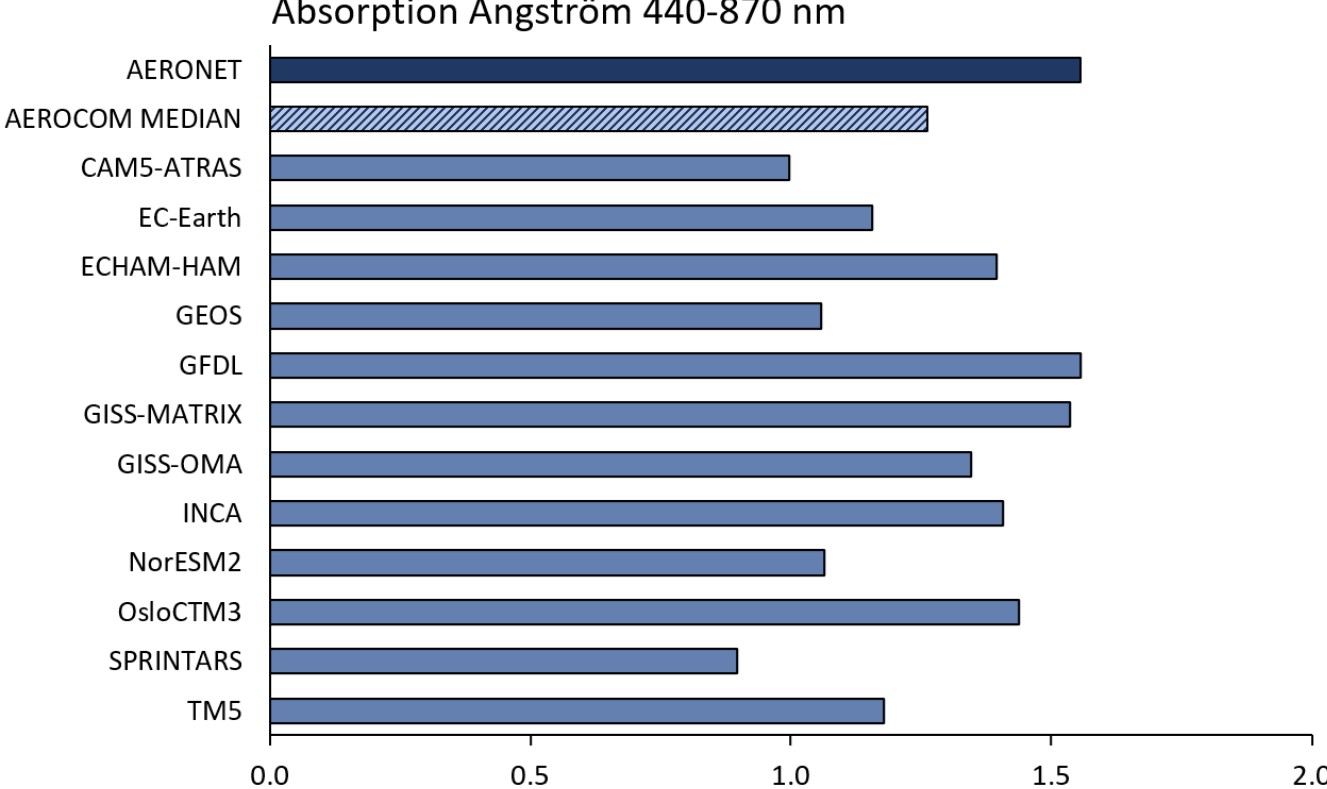

**Figure 12: Aerosol absorption Ångström exponent based on total AAOD at λ = 440 nm and λ = 870 nm calculated from monthly means requiring 25% daily coverage to compute AERONET monthly means from daily values. For GISS-OMA and GISS-MATRIX the AAE was calculated based on λ = 550 nm and λ = 870 nm.**

Figure 12 shows the aerosol absorption Ångström exponent (AAE) which expresses the spectral dependence of AAOD. The AERONET AAE is computed from a retrieval of a size distribution and complex refractive index that is constrained by direct sun radiance measurements. The AAE in the AeroCom models has been calculated with AAOD at λ = 440 nm and λ = 870 nm (see Methods) in the nearest grid cell to AERONET stations requiring 25% daily coverage (i.e., at least 7 days) to compute AERONET monthly means from daily values. The spectral dependence varies quite substantially between the models ranging from 0.9 to 1.6, with an average 1.3. AAE from the AERONET sites is 1.6. Since BC particles are small (less than 50 nm) with wavelength-independent index of refraction over the visible spectrum, AAE is expected to be 1 for externally mixed BC, but this may not be true for internally mixed, aged BC (Bergstrom et al., 2002; Schuster et al., 2016). Organic aerosols' MAC decreases sharply with wavelength and the AAE is shown to be larger than 1 (Olson et al., 2015). For dust particles AAE is suggested to be larger than 1, but the uncertainties are larger compared to BC (Samset et al., 2018; Linke et al., 2006). Schuster et al. (2016) argue that it is difficult to separate AAE of dust and BC/OA, because AAE is also affected by size and published values of AAE of pure dust vary from less than 0 to larger than 3 depending on the relative fractions of hematite and goethite.





Figure 13 shows the AAE split into BC, OA, and dust for the five models (CAM5-ATRAS, ECHAM-HAM, GFDL, INCA
and OsloCTM3) with absorption per species at $\lambda$ = 440 nm and $\lambda$ = 870 nm. BC is around 1 (0.9-1.3), dust is around 2 (2.0-
2.2), while OA is much lower than 1 (0.3-1.0), except for one model (OsloCTM3) which has a AAE for OA 16.1. This is
because the absorption for OA near 870 nm is close to zero in this model. OA has stronger spectral dependence compared to
BC (see Fig 1), which enhances the absorption at shorter wavelengths. Given equal particle sizes, AAE for OA will therefore
be larger than for BC. However, Fig. 10 shows that the spectral dependence for OA in these models (except OsloCTM3) is
weak. This contrasts with observations, both from laboratory studies and over observational sites, which finds stronger spectral
dependence for OA than BC (e.g., Bond, 2001; Kirchstetter et al., 2004; Schnaiter et al., 2006). Many of the AeroCom models
have not updated their OA refractive indices to include BrC. BrC is mostly responsible for the spectral dependence.

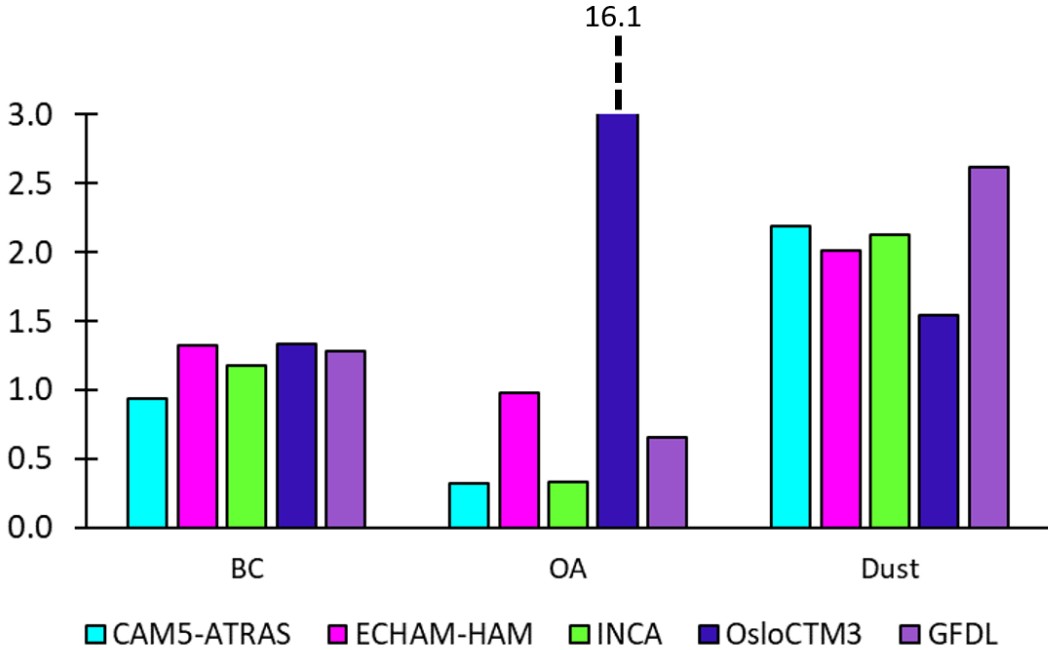


**Figure 13: Global mean aerosol absorption Angstrom exponent based on total AAOD at $\lambda$ = 440 nm and $\lambda$ = 870 nm split into BC,**
**OA, and dust.**
**3.5 Anthropogenic AAOD**
Figure 14 shows the anthropogenic total AAOD, here defined as changes in total AAOD between 1850 and 2010 for the 11
models reporting AAOD for 1850. The global mean total AAOD change 2010 - 1850 is 0.0024 [0.0008 – 0.0047]. The
geographical pattern is very similar among the models, which is expected due to their similar changes in anthropogenic and
biomass burning emissions. However, the spread in global mean numbers is quite high (1.7). Global mean total AAOD in 1850





is 0.003 [range 0.0012 to 0.0065], and the spread is quite high (1.8 and standard deviation; 0.0015) (Figure S9 in Supplement).
This means that some part of the variability between the AeroCom models can be attributed to preindustrial/natural aerosols.

**Figure 14: Change in total AAOD λ = 550 nm between 1850 and 2010 from the models; (a) annual global mean, (b) annual zonal mean (c) the global seasonal cycle and (d) annual mean spatial distributions.**



## 4 Summary and discussion

15 different aerosol models from AeroCom Phase III have reported total aerosol absorption optical depth (AAOD) and for the first time 11 (10) these models have reported in a consistent experiment the contributions to AAOD from BC, dust and organic aerosol.

- The global model mean (median) total AAOD is 0.0056 (0.0055), which is 31% higher than in AeroCom Phase II, but within one standard deviation. The models show a maximum in areas with biomass burning, over large industrial areas and over the Sahara Desert. Compared to retrieved AAOD from AERONET stations, the models yield lower absorption. The AERONET mean AAOD is 0.046 while the AeroCom model mean is 0.035 (range 0.015-0.077) at these selected AERONET sites.  For comparison, the global mean total AOD (absorption + scattering) for the same models is 0.129 [range 0.097 – 0.156]. The correlation between global mean AAOD and AOD is 0.6.
- The anthropogenic total AAOD (changes in AAOD between 1850 and 2010) is 0.0024, which is 42% the total AAOD.
- The spectral dependence varies substantially between models. The multi-site averaged AAE from the AERONET sites is 1.6 while the respective averages for the individual models range from 0.9 to 1.5.
- The models that report absorption per species yield AAOD contributions of 58% due to BC [range of 34% to 84%], 28% [12 - 44]% due to dust and 14% [4 - 49]% due to OA (average contribution). Models with the lowest BC absorption have the highest OA absorption, illustrating the complexities in separating the species and mixing assumptions in models where internal mixtures are assumed depending on how BC AAOD is calculated. However, the absorption of BC and OA is not additive (Fig 5). The total AAOD is less variable (spread 1.4) than BC AAOD and OA AAOD (both has spread 2.5).
- The global model mean (median) BC AAOD is 0.0028 (0.0021) [range 0.0007 - 0.0077]. The seasonal cycle follows the biomass burning season in Africa and South America. The model annual mean BC MAC value is 8.6 m$^2$ g$^{-1}$ a [3.1 – 15.0] m$^2$ g$^{-1}$. Near-surface observations of BC MAC values 550 nm from various locations vary between 5.7 up to 20.0 with an average of 10.9 m$^2$ g$^{-1}$ and a standard deviation of 3.1 m$^2$ g$^{-1}$.
- Globally averaged dust AAOD at 550 nm is approximately half that of BC (dust AAOD peaks for lower wavelengths). The global model mean (median) dust AAOD is 0.0013 (0.0011) [range 0.0006 to 0.0021].
- The global model mean (median) OA AAOD is 0.0005 (0.0004) [range 0.0002 to 0.0016]. Of the five models which reported OA absorption for 440 and 870 nm, four of them show very weak spectral dependence, in contrast to observations. We recommend the AeroCom models to update their OA refractive indices based on available measurements.

The AeroCom models have similar BC emissions, but we still find a substantial spread in BC absorption. This can be explained by a relatively large variability in both BC lifetime (ranging from 4 to 9 days) and the vertical distribution in the atmosphere. The lifetime and mixing state are coupled, as enhanced mixing reduces lifetime (Stier et al. 2006). Different aerosol mixing





assumptions and the associated optical calculations in the models add to the uncertainties in absorption. Some models use Maxwell-Garnett mixing rules (INCA, NorESM2, TM5), some use volume averaging (ECHAM-HAM, ECHAM-SALSA), while others use a core-shell mixing (CAM5-ATRAS). Stier et al. (2007) compared different mixing rules using a consistent setup in one single model (ECHAM5- HAM) and found a moderate influence of the mixing rules (10%). This was found to be weaker than the uncertainties in the imaginary index. We also find very little correlation between the imaginary index and mass absorption coefficients. For BC just three different refractive indices are used by the models, while the spread is not related to this choice. There are also differences in how models with internal mixing diagnose the aerosol species absorption contributions. Some models calculate component absorption by differences between simulations with and without the specific component included (CAM-ATRAS), while others use volume weighting, either by the relative volume of each component in the mixture (GFDL) or by volumes at size-bin-level (NorESM2). It should be noted that this issue is related to the separation of aerosol radiative properties into individual components and does not affect the actual radiative aerosol properties applied in the models forcing calculation. We recommend that the role of size and mixing rules and diagnostic procedures should be investigated in more detail to understand the differences in mass absorption coefficients.

Schulz et al. (2006) calculated the normalized BC RF per BC AAOD for AeroCom Phase I (model average 153 with standard deviation 64). Using these numbers combined with our estimates for mean BC AAOD 2010-1850 (0.002) yields a BC RF of 0.30 Wm$^{-2}$ with a standard deviation 0.25. A better understanding of the processes and properties of absorbing aerosols is critical to reduce the large uncertainties in aerosol-climate interactions. In particular, we have found that the imaginary indices are not explaining much of the AAOD variance, except slightly for dust. We suggest that the optical calculations need more testing e.g., in a box model, or by exchanging optical calculations among models.

**Code and data availability** All data used in this study are stored on servers of the Norwegian Meteorological Institute and can be provided upon request. All analysis scripts (using IDL and python) are stored at CICERO servers and can be provided upon request.

**Author contribution**

MS and BHS designed the study. MS did most of the analysis and wrote most of the paper. JG provided data and scripts for the AERONET comparisons and the AEROCOM-MEDIAN fields. CWS provided measurements values of BC MAC. The other co-authors provided model data. All co-authors provided feedback to the paper.

**Competing interests**

The authors declare that they have no conflict of interest.

**Acknowledgements**



MS, BHS., CWS. and MTL acknowledge funding from the Research Council of Norway, through grant nr. 244141 (NetBC)
and grant nr. 248834 (QUISARC). RC-G, AK., MS were supported by the European Union's Horizon 2020 grant agreement
No 641816 (CRESCENDO). H.M. was supported by the Ministry of Education, Culture, Sports, Science and Technology of
Japan and the Japan Society for the Promotion of Science (MEXT/JSPS) KAKENHI Grant Numbers JP17H04709,
JP19H05699, and JP20H00638, MEXT Arctic Challenge for Sustainability II (ArCS-II) project (JPMXD1420318865), and
the Environment Research and Technology Development Fund 2–2003 (JPMEERF20202003) of the Environmental
Restoration and Conservation Agency. TT was supported by the NEC SX supercomputer system of the National Institute for
Environmental Studies, Japan, the Environment Research and Technology Development Fund (grant no. JPMEERF20202F01)
of the Environmental Restoration and Conservation Agency, Japan, and the Japan Society for the Promotion of Science (JSPS)
KAKENHI (grant no. JP19H05669). PS acknowledges support from the European Research Council (ERC) project RECAP
under the European Union's Horizon 2020 research and innovation programme with grant agreement 724602 and from the UK
Natural Environment Research Council project NE/P013406/1 (A-CURE). MS and AK acknowledge funding from the
European Union's Horizon 2020 Research and Innovation programme, project FORCeS, under grant agreement no. 821205,
by the Research Council of Norway INES (grant no. 270061), and KeyClim (grant no. 295046)). High performance computing
and storage resources were provided by the Norwegian Infrastructure for Computational Science (through projects NN2345K,
NN9560K, NS2345K, and NS9560K). The AeroCom database is maintained by the computing infrastructure efforts provided
by the Norwegian Meteorological Institute.

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
