# Peer review of "Aerosol absorption in global models from AeroCom Phase III"

_Atmospheric Chemistry and Physics, 2021_

## Referee Comment (RC2)

This paper presents climatologies of aerosol absorption for 15 Aerocom models. The authors discuss the spread of the modeled results, and they compare modeled AAOD and AAE to AERONET. They find that modeled AAOD is biased low of AERONET, which has been a consistent finding for the lifetime of AERONET (e.g., *Sato et al.*, 2003; *Koch et al.*, 2009; *Bond et al.*, 2013). The paper is a light read and will probably be cited by the modeling community, but it does not provide the reader with new insight about why the model diversity exists, or why model AAODs continue to be biased low of AERONET AAODs (even 18 years after this was first noticed by *Sato et al.*, 2003).

The overarching goal of the paper is presented by the authors on Line 105:

*"We aim to better quantify the sources of model spread by separating absorption per species (BC, OA, and dust) and investigate regional and seasonal differences."*

Unfortunately, I do not believe that they accomplished this. They do separate absorption by species, but they do not use the same technique to compute species-specific absorption for the various models, and seemingly leave it up to the various modelers to submit their favorite technique for this comparison. This makes it very difficult for the reader to sort out the causes of the model spread that are presented in the paper, and indeed, the authors themselves seem to give up, attributing the model spread in MAC and AAOD on line 369 to *"...the complexities in separating the species and mixing assumptions where internal mixtures are assumed depending on how BC AAOD is calculated"*.

The authors recognize that computing species-specific absorption is easy for external mixtures, but at least nine of the models in Table 2 include internal mixing. Table 2 provides 1-2 sentences about how the models with internal mixing partition absorption amongst the individual species, but the descriptions are too brief for the reader to clearly understand what is being done. The main text does not help much, either. In the summary, the authors conclude on line 398 with: *"We recommend that the role of size and mixing rules and diagnostic procedures should be investigated in more detail to understand the differences in mass absorption coefficients."*. This leaves the reader wondering, though – why didn't the authors do that themselves already in this paper?

It would not be that difficult of a task to understand the spread in the MACs. The mass absorption coefficient (MAC) at any given wavelength for a pure species (i.e., external mix) is a function of the complex refractive index (mostly the imaginary component), size distribution, and density. For internally mixed aerosols, the volume fraction of the absorbing inclusions also plays a role. Thus, only 3 parameters are needed to analyze the MAC of external mixtures, and 4 paramaters are needed for internal mixtures. A 5th parameter that would be useful for analyzing the spread in species-specific column AAOD is column mass loading, but this is never mentioned in the article. The paper has the potential to make a significant contribution if the authors broke down the analysis into these 4-5 components. It would also be helpful to analyze the external and internal mixtures separately.

**How to attribute absorption to different species for internal mixtures**

The authors state that it is conceptually difficult to report separate absorption by species for internal mixtures (lines 85, 204, and 267). The conceptual difficulty occurs because the sum of the absorptions of the component species in an internal mixture does not equal the absorption of an internal mixture. The simplest example is a black carbon core embedded in a non-absorbing shell (like water) – the core-shell particle will have much greater absorption than the core in isolation. However, the carbon core is still responsible for all of the absorption in a core-shell particle, as the water is not absorbing any photons.

Thus, the conceptual difficulty of attributing absorption to different species in an internal mixture should not be that difficult – we can compute the effect of any absorber in a model by removing it. For example, when we compute direct aerosol radiative effects, we compute the difference between "with aerosols" and "without aerosols." Likewise, direct aerosol forcing is computed by differencing the present-day aerosols with an estimate of the aerosol loading in some reference year (e.g., 1850). Why not do the same thing with individual aerosol species? Thus, the AAOD effect caused by a target species is: (total aerosol absorption) - (aerosol absorption in the absence of the target species). The resulting AAOD represents the physical impact of omitting the target species from the atmosphere, which is what we are seeking. Glancing through Table 2, it seems that CAM5-ATRAS is the only model that got this right (good job CAM5-ATRAS!).

Some of the AeroCom modelers have lamented in the past that this method of speciating absorption alters the aerosol size distribution, but so what? Our traditional computations of DARE and DARF dramatically alter the aerosol size distribution, too. In the case of DARE, after all, it totally removes the size distribution! Why is it folly to remove part of a size distribution instead of all of it? The other definitions in the table, including the old Aerocom protocol* recommendation at https://aerocom.met.no/protocol_expl.html, do not represent anything physical regarding internal mixtures and do not recognize that the absorption of an internal mixture is often greater than the sum of the absorbing components. In addition, the AeroCom protocol prescribes a set of densities that may or may not be consistent with any particular model. Thus, it seems as though this protocol was created mainly because of its simplicity.

*Aerocom protocol at https://aerocom.met.no/protocol_expl.html:
Aerosol optical depth per species for aerosol internal mixtures: A check on how you compute the contribution of a given aerosol species to total aerosol optical depth if you have internal mixtures in the model: Procedure recommended during AEROCOM workshop: Compute volume fraction of aerosol species in aerosol particle volume (without water!!) and retrieve with that fraction the aerosol optical depth for a given species. Apply the following densities for the major species (Dust = 2650 kg/m3 / Sea salt = 1600 / Sulfate = 1769 / Black Carbon = 1500 / Particulate organic matter = 1500 )

Given the wide variety of methods for attributing AAOD to the absorbing species in this

paper, the values for the component AAODs in Table 3 are not comparable between models. Since optical properties are computed offline, it would not be that difficult for each modeler to compute component AAODs with a single sensible protocol (ideally, as I described above for the internal mixtures) with a single set of refractive indices and densities for the absorbing aerosols. Constraining all of the models in the same way like this would make this paper strong and valuable. Otherwise, the paper just reports what we basically already know – the models are different from each other and the observations, and we have not quantified why.

**Table 2**

I really like the idea of Table 2 because it is a great idea to have the mixing assumptions for the AeroCom models all in one place. I don't understand some of the methods and terminology, though. For instance, what does *"Core-shell for internally-mixed BC particles; Volume mixing for pure BC and BC free particles"* mean for the CAM5-ATRAS model? A reader might surmise from this text that the authors are using volume-averaged refractive indices for external mixtures, which does not make sense.

It is also not clear which components are being internally mixed in many of the models. For example, ECHAM-HAM, ECHAM-SALSA, and GISS-MATRIX do not indicate which absorbing species they are treating as internal mixtures (BC, OA, dust, or some combination of all 3).

I was surprised to see that the EC-Earth3 model treats sulfate, ammonium-nitrate, organic aerosols, sea salt, and water as homogenous mixtures described by the Bruggeman mixing rule. This rule is usually applied to dry insoluble components, like mineral mixtures. The table seems to indicate that all of these components are mixed together into one internal mixture, but is that accurate?

The SPRINTARS model describes internal mixing of BC with OC in the 2nd column of Table 2, but the third column says that the BC AAOD is calculated assuming that all BC is externally mixed. Thus, this is inconsistent.

The NorESM2 model states:
*Internal and external mixing. Maxwell-Garnett is used for calculation of refractive index of internal mixing of BC with other components, otherwise volume mixing.*

What is volume mixing? If the modeling is applying volume averaging of refractive indices to external mixtures, then this is not the correct way to compute optical properties of external mixtures. Thus, "volume mixing" for external mixtures needs to be defined.

**Figures**

**Figure 1** is based upon Mie calculations taken from *Samset et al.* (2018). However, *Samset et al.* (2018) provides information about the size distributions, refractive indices, and densities that are required to make this figure, whereas this article does not. Additionally, this article fails to explain that the *Bond and Bergstrom* (2006) recommended value of MAC = 7.5 m2/g can not be achieved using the *Bond and Bergstrom* (2006) recommended refractive indices and densities, which explains why the "fresh" symbols lie outside of the shaded region. Similarly, the "coated" and "collapsed, uncoated" symbols are left unexplained in this figure. *Samset et al.* (2018) explains that the symbols are based upon values found in the literature and lists several citations.

These are important details because a reader should be able to understand all figures on the basis of the material in the article. Additionally, since the caption states that MAC is separable amongst the species in Figure 1, the reader needs to know the range of size distributions and densities associated with the shaded regions.

**Figure 2:** Given that this paper is about modeling aerosol absorption, some discussion about the use of OPAC optical properties is warranted. Six of the 14 models in Figure 2 use BC imaginary refractive index (BC IRI) of 0.44, which is the value given by OPAC. That means that nearly 1/2 of the models are still using OPAC, but more than 1/2 of the models have figured out a way to move beyond OPAC. Way back in 2006, *Bond and Bergstrom* (2006) reported this in their assessment of BC refractive indices: "The value commonly used by climate modelers ($m = 1.74 - 0.44i$ at 550 nm) represents none of the possible refractive indices and should be retired."

*Bond and Bergstrom* (2006) also said:
"The history of refractive index values tabulated by Shettle and Fenn (1976, 1979) is worth special mention. These are by far the most prevalent values for use in climate modeling, and have been incorporated into widely-cited literature, including a book by d'Almeida et al. (1991), and the Optical Properties of Aerosols and Clouds (OPAC) program (Hess et al. 1998). The original work by Shettle and Fenn (1976) averaged values from an earlier review by Twitty andWeinman (1971). In turn, the averaged data are taken from McCartney et al. (1965), who measured three coals, and Senftleben and Benedict (1918), who reported soot generated from an arc lamp. The review does not incorporate most of the findings on soot in the combustion literature, and indeed was written before most of that work was available. The precision of both n and k provided in OPAC values (three decimal places) is unwarranted, given this history. The OPAC value of $1.74 - 0.44i$ is drawn from incompletely graphitized carbon and has a lower value of k than most soot...
...The optical and physical data for LAC propagated by d'Almeida et al. (1991) have some interesting properties. Along with an imaginary refractive index that is too low, these authors recommend: (1) a particle size that is far too small (23 nm is the approximate size of primary spherules, not aggregates); (2) a geometric standard deviation that is somewhat too large (2.0);

and (3) a density that is far too low (a density of 1.0 is never observed; Fuller et al. (1999) tabulate measurements indicating densities of about 1.8 g/cm3). Despite returning to the string of citations that led to d'Almeida et al. (1991), we have been unable to unearth the sources of these values. When compared with measured values, each of the individual assumptions above may lead to an error of 50-75% in calculated properties that affect climate forcing."

It is unfortunate that so many models did not follow the *Bond and Bergstrom* (2006) recommendations. Since the present article is about model diversity of absorption, this would clearly be a great place to revive the OPAC BC issue.

**Comparisons to AERONET data**

Given the text on lines 192-200, does it even make sense to compare any extrinsic AERONET properties (like AAOD) to the modeled values, as in Figure 4?? That is, the AERONET dataset that the authors are using require $AOD(440) > 0.4$ (per line 193), but the models use all available AOD (per line 195). So why would there be an expectation that any modeled extrinsic parameter would compare well with AERONET under these conditions (even with a high-resolution grid)? If the authors checked the coincident-only AODs that are obtained during the same scans that produce the AAODs, I expect that they would see a significant bias in AOD for the same reason – modeled extrinsic parameters will not compare well to AERONET when you discard low-load cases in AERONET but keep low-load cases in the model output. (If they don't see a bias, that means that they are not using enough AERONET sites with $AOD(440) < 0.4$, or something is very wrong with the model). In fact, model comparisons to AERONET should be **required** to show coincident-AOD comparisons whenever there is a comparison of extrinsic parameters (like AAOD).

I suggest the following. . . plot aerosol co-albedo (absorption/extinction) instead of AAOD. Co-albedo is a pseudo-intrinsic parameter that is much less susceptible to column aerosol loading than AAOD.

**Unclear Definitions for Organic Aerosol, Organic Carbon, and Brown Carbon**

The authors are not clear about the differences between OA (organic aerosol), organic carbon, and brown carbon. Indeed, the text doesn't seem to recognize that BrC is a subset of OC and that OC is a subset of OA. On line 66, the authors refer to BrC/OA, implying that BrC and OA are one and the same species. On line 117 and in Table 2, though, the authors discuss OA/OC ratios, which are always greater than 1. On line 124, the authors mention that the *"OsloCTM3 model divide OA into into a mix of absorbing and non-absorbing species. . . "*.

The authors need to inform their readers that the brown shaded regions in Figure 1 correspond to organic aerosols that are "washed" with solvents like acetone and methanol in

order to extract the absorbing organic aerosols from the non-absorbing organic aerosols. The OsloCTM3 model seems to recognize this by separating absorbing and non-absorbing OC. The other models apparently use a single set of optical properties that encompasses all organics (both absorbing and non-absorbing), so they should be using MACs that are much lower than the values shown in Figure 1. Additionally, BrC is produced by hulis and is largely found in biomass burning and is not observed in fossil fuel burning. Thus, the models should be using a different set of optical properties for OC that originate from modern urban areas than the optical properties that they use for biomass burning. None of these concepts are clearly recognized or discussed in the text.

**Wavelength Dependence of Absorption**

Section 3.4 and Figure 11 are another missed opportunity for some good discussion. All of the models except for OsloCTM3 show significant OA absorption at 870 nm, but Figure 1 indicates no detectable BrC absorption at this wavelength. Clearly, none of the models are using the recent measurements for BrC that are shown in Figure 1 (except perhaps OsloCTM3 – good job Oslo!). This begs the question – what are the models using for OA? Given that this is a paper on understanding the model spread of AAOD, this is an important topic. Sure, the authors provide OA refractive indices at 550 nm in Figure 2, but what measurements are the basis of the refractive indices in the models, and why are the modeled OA MACs and dust MACs buried in the supplement? The OA MACs in Fig S8 are on the low end of Fig 5 (assuming that the wavelength in S8 is 550 nm, but that is not stated), so does that mean that the OA imaginary refractive indices are more or less constant wrt wavelength? Also, the OA MAC varies by a factor of 4+ in Figure S8 – how come this huge range of modeled optical properties for OA is not discussed in the text? Same question for the MAC of dust, which varies by a factor of 3+ in Figure S8.

**Going Forward**

There is potential for an excellent paper in this material. I would like to see:

- An analysis of the models with external mixing. What does the spread in MACs and AAODs for this subset of models look like, and what is the cause of the spread? Size, refractive index, and density are the important parameters. Analyzing MAC wrt IRI/density will probably be enlightening. Column mass loading of the absorbing aerosols should be a factor.

- A single optics module applied to the models with internal mixing, as much as possible. Use the CAM5-ATRAS method for splitting absorption amongst components and analyze the spread in MACs and AAODs, and abandon the old AeroCom protocol.

Pay attention to the mass/volume fraction of absorbers in the mixtures, especially for BC. Here again, column mass loading will be important for understanding AAOD.

- With separate analyses of the external and internal mixtures now in hand, how do these two analyses compare to each other? How do they compare to the models with internal mixing that are not amenable to the approach in the 2nd bullet?

- Don't compare to any extrinsic AERONET parameters, since the $AOT(440) > 0.4$ restriction skews the AERONET data to large aerosol loadings. Use AERONET's single-scatter co-albedo, or omit.

- Pay attention to BC density. The use of OPAC BC with a realistic density (as opposed to OPAC's recommended value of 1 m2/g) is probably the cause of the lowest MACs in Figure 9. This could be easily remedied by abandoning OPAC. The densities of all absorbers should be included in a table.

- Abandon OPAC. Since the authors are only considering a few wavelengths (440, 550, and 870 nm), abandoning OPAC is not a tall order.

- Inform the reader about where each model obtains their information for optical properties of aerosols (i.e., provide citations). Many of them use OPAC, but where do the other models find complex refractive index, size, and density information? This will allow the reader to understand which models are using the latest measurements, and which ones are lagging.

- A description of the size distributions that the models use is needed. Lognormal radii, widths, etc., should be tabulated.

- Make sure that the figures are self-contained (in the sense that the reader does not have to go to another paper to understand how they were created).

- Clear up the phraseology everywhere, especially in Table 2 where it is so important. Make sure that all text is clear so that a reader understands the intent of your words without ambiguity. There are many co-authors on this paper for proofreading, so this should not be an issue.

- Clearly define OA, OC, and BrC early in the text.

All of this should be do-able with the model runs already utilized in this paper.

**Line-by-line issues that need to be addressed:**

Line 70:
*"While fine-dust particles mostly scatter solar radiation, coarse dust also absorbs moderately in the visible and near-infra-red spectrum."*

I would like to see a citation for this, because I am not sure if it is accurate. Iron oxides in dust are small, so I don't see why they would not be present in the fine-mode dust as well as coarse mode dust.

Line 157:
"...their imaginary parts of the refractive index vary a lot ($1.75 + 0.44i$ for SPRINTARS and $1.85 + 0.71i$ for GISS-MATRIX (Fig. 2)."

This is a missed opportunity – why not use the same BC refractive index and BC density in all of the models? This would eliminate a significant source of diversity so that part of the analysis could be focused on the remaining causes of diversity (i.e., size distribution and column mass loading). Additionally, the diversity of BC density is entirely missing from this paper. BC density is very important because it is inversely proportional to MAC (all else being equal), and therefore has a direct impact on both MAC and the AAOD computed by the models. Finally, why not compare column mass loading amongst the models as well? I believe that we could learn something by studying mass diversity alongside the MAC, AOD, and AAOD diversity, but column mass loading was not even mentioned as a source of diversity in this article.

Line 199:
"However, using a high-resolution simulation of global aerosols, Schutgens (2020) found a much smaller bias of 9%"

Ending the paragraph of caveats with this sentence is a bit of a hoodwink, as the present study does not use a high-resolution model.

Line 213:
"In NorESM2 the additional absorption is from sea-salt and sulfate (mixed with BC, dust and OA). In GFDL BC is internally mixed with SO4, so the additional absorption is due to SO4 (mixed with BC)..."

This is not explained very well because the most important part of the explanation is in parentheses. Reading this, one might think that sea-salt, sulfate, etc. are absorbing photons. The authors should clearly point out that internally mixed BC has greater absorption than externally mixed BC, and that this is because the internal mixtures have greater geometrical cross-sectional areas than the BC inclusions within the mixtures. The sea-salt and sulfate in the mixtures are not absorbing any photons, whether they are hosting absorbing inclusions or not. This needs to be clear.

Line 323:
The authors mention BC particle size as 50 nm, but they do not state whether this is radius or diameter. Additionally, 50 nm is closer to the diameter of a spherule than a BC aggregate. For example, *Schwarz et al.* (2008) measured size distributions of BC with an SP2, and they obtained a median diameter of ~200 nm.

Line 363: If the global mean modeled AOD is 0.129 and the global model mean AAOD = 0.035 (per line 188), then the global mean SSA should be close to (1 - .035/.129) = 0.73. Why is the global mean AAOD so high wrt to the global mean AOD?

Line 391-393:
*"We also find very little correlation between the imaginary index and mass absorption coefficients. For BC just three different refractive indices are used by the models, while the spread is not related to this choice."*

This sentence appears in the Summary, but I don't see IRI/MAC correlations discussed in the paper. However, it is not surprising that the authors do not see a relationship between IRI and MAC if they did not account for the different densities used in the models. For instance, OPAC uses a density of 1 g/cm3, which results in a MAC of ∼10m2/g (per *Bond and Bergstrom*, 2006). A more realistic choice of 1.8 g/cm3 for BC density would reduce the MAC to 10/1.8 = 5.5 m2/g. The imaginary refractive index makes a huge difference to the BC MAC, but not if one simultaneously tunes the BC density.

Line 398:
*"We recommend that the role of size and mixing rules and diagnostic procedures should be investigated in more detail to understand the differences in mass absorption coefficients."*

I agree completely, but why didn't the authors do that in this paper? That is actually what many readers will be looking for in this article.

Line 403:
*"In particular, we have found that the imaginary indices are not explaining much of the AAOD variance, except slightly for dust."*

Here again, I did not see this discussed at all in the paper. The final section should be a summary of the details that are presented in the paper, not the introduction of a new result.

**Minor issues:**
Line 51:
*"The three absorbing species are rarely observed as single species,..."*
This sentence does not make sense to me...Shouldn't this go unstated by definition?

Line 58:
*"but these calculations are approximate (using mixing rules or the assumptions of a co-centric core/shell structure)..."*
This sentence seems to imply that internal mixing requires additional significant assumptions wrt external mixing, which is not the case. Core/shell computations are exact, although the core/shell structure is an approximation for the shape of aerosols in the atmosphere. Likewise, Mie Theory is also exact for the spherical particle approximation that is used for all externally-mixed aerosols in the models (with the exception that some models use a spheroidal approximations for dust), but spheres and spheroids are still approximations for

particle shape. Thus, approximations associated with shape are required for both internal and external mixing. Finally, errors associated with effective medium approximations have been tested by many authors (e.g., *Martins et al.*, 1998; *Fuller et al.*, 1999; *Lesins et al.*, 2002) and are likely swamped by other modeling errors associated with mixing fractions and assumed component refractive indices.

Line 61: Begins with "However,..." and then essentially repeats the information on line 58. Sentence should start with a different word.

Line 102: "the separation into fine ($< 1\mu$m) and coarse mode ($> 1\mu$m) AOD..." Presumably the authors mean diameter, but this should be specified.

Line 119: "and 11 models have provided absorption split into BC and dust (OA)." Why is OA in parentheses?

Line 147: Authors are using Version 2 AERONET, but which level? That is, Level 1.5, or 2.0?

Line 183 (Fig 4) and Line 185: The authors need to specify whether they are using Version 2 or Version 3 AERONET and Level 1.5 or Level 2, here. The authors mention Level 2 later on Line 193, but it is not clear whether they realize that Version 2 means something different than Level 2.

Line 228: Table 2 says that Sprintars has internal mixtures of BC with OA. Why do the authors include it as a model with external mixing?

Line 276: Replace "Methods" with "Section 2 – Methods" for improved reader navigation.

Line 325 & 328: BC AAE was addressed in *Schuster et al.* (2016b), not *Schuster et al.* (2016a).

Line 335: Should be Fig 13, not Fig 10.

Line 360:
*"Compared to retrieved AAOD from AERONET stations, the models yield lower absorption."*

One would hope so, since the AERONET dataset that the authors use requires $AOD(440) > 0.4$, whereas the models are not using the same restriction.

Line 381: It is good that that the authors are recommending and update for modeled OA, but it would be nice if they reported what the models are actually using right now, too.

**References**

Bond, T., and R. Bergstrom (2006), Light absorption by carbonaceous particles: An investigative review, *Aerosol Sci. Technol.*, *40*(1), 27–67.

Bond, T., et al. (2013), Bounding the role of black carbon in the climate system: A scientific assessment, *J. Geophys. Res.*, *118*, 1–173, doi:10.1002/jgrd.50171.

Fuller, K., W. Malm, and S. Kreidenweis (1999), Effects of mixing on extinction by carbonaceous particles, *J. Geophys. Res.*, *104*(D13), 15,941–15,954.

Koch, D., et al. (2009), Evaluation of black carbon estimations in global aerosol models, *Atmos. Chem. Phys.*, *9*, 9001–9026.

Lesins, G., P. Chylek, and U. Lohmann (2002), A study of internal and external mixing scenarios and its effect on aerosol optical properties and direct radiative forcing, *J. Geophys. Res.*, *107*(D10), 4094, doi:10.1029/2001JD000973.

Martins, J., P. Artaxo, C. Liousse, J. Reid, P. Hobbs, and Y. Kaufman (1998), Effects of black carbon content, particle size, and mixing on light absorption by aerosols from biomass burning in Brazil, *J. Geophys. Res.*, *103*(D4), 32,041–32,050.

Samset, B., C. Stjern, E. Andrews, R. Kahn, G. Myhre, M. Schulz, and G. Schuster (2018), Aerosol Absorption: Progress Towards Global and Regional Constraints, *Curr Clim Change Rep*, *4*(65), doi:10.1007/s40641-018-0091-4.

Sato, M., J. Hansen, D. Koch, A. Lacis, R. Ruedy, O. Dubovik, B. Holben, M. Chin, and T. Novakov (2003), Global atmospheric black carbon inferred from AERONET, *Proc. Natl. Acad. Sci.*, *100*(11), 6319–6324.

Schuster, G., O. Dubovik, and A. Arola (2016a), Remote sensing of soot carbon – Part 1: Distinguishing different absorbing aerosol species, *Atmos. Chem. Phys.*, *16*, 1565–1585, doi:10.5194/acp-16-1565-2016.

Schuster, G., O. Dubovik, A. Arola, T. Eck, and B. Holben (2016b), Remote sensing of soot carbon – Part 2: Understanding the absorption Ångström exponent, *Atmos. Chem. Phys.*, *16*, 1587–1602, doi:10.5194/acp-16-1587-2016.

Schwarz, J., et al. (2008), Measurement of the mixing state, mass, and optical size of individual black carbon particles in urban and biomass burning emissions, *Geophys. Res. Lett.*, *35*, L13810, doi:10.1029/2008GL033968.

---

## Author Comment (AC1)

This paper presents climatologies of aerosol absorption for 15 Aerocom models. The authors discuss the spread of the modeled results, and they compare modeled AAOD and AAE to AERONET. They find that modeled AAOD is biased low of AERONET, which has been a consistent finding for the lifetime of AERONET (e.g., Sato et al., 2003; Koch et al., 2009; Bond et al., 2013). The paper is a light read and will probably be cited by the modeling community, but it does not provide the reader with new insight about why the model diversity exists, or why model AAODs continue to be biased low of AERONET AAODs (even 18 years after this was first noticed by Sato et al., 2003). The overarching goal of the paper is presented by the authors on Line 105: "We aim to better quantify the sources of model spread by separating absorption per species (BC, OA, and dust) and investigate regional and seasonal differences."

Unfortunately, I do not believe that they accomplished this. They do separate absorption by species, but they do not use the same technique to compute species-specific absorption for the various models, and seemingly leave it up to the various modelers to submit their favorite technique for this comparison. This makes it very difficult for the reader to sort out the causes of the model spread that are presented in the paper, and indeed, the authors themselves seem to give up, attributing the model spread in MAC and AAOD on line 369 to ". . . the complexities in separating the species and mixing assumptions where internal mixtures are assumed depending on how BC AAOD is calculated".

The authors recognize that computing species-specific absorption is easy for external mixtures, but at least nine of the models in Table 2 include internal mixing. Table 2 provides 1-2 sentences about how the models with internal mixing partition absorption amongst the individual species, but the descriptions are too brief for the reader to clearly understand what is being done. The main text does not help much, either. In the summary, the authors conclude on line 398 with: \We recommend that the role of size and mixing rules and diagnostic procedures should be investigated in more detail to understand the differences in mass absorption coefficients.". This leaves the reader wondering, though - why didn't the authors do that themselves already in this paper?

It would not be that difficult of a task to understand the spread in the MACs. The mass absorption coefficient (MAC) at any given wavelength for a pure species (i.e., external mix) is a function of the complex refractive index (mostly the imaginary component), size distribution, and density. For internally mixed aerosols, the volume fraction of the absorbing inclusions also plays a role. Thus, only 3 parameters are needed to analyze the MAC of external mixtures, and 4 paramaters are needed for internal mixtures. A 5th parameter that would be useful for analyzing the spread in species-specific column AAOD is column mass loading, but this is never mentioned in the article. The paper has the potential to make a significant contribution if the authors broke down the analysis into these 4-5 components. It would also be helpful to analyze the external and internal mixtures separately.

**How to attribute absorption to different species for internal mixtures**
The authors state that it is conceptually difficult to report separate absorption by species for internal mixtures (lines 85, 204, and 267). The conceptual difficulty occurs because the sum of the absorptions of the component species in an internal mixture does not equal the absorption of an internal mixture. The simplest example is a black carbon core embedded in non-absorbing shell (like water) { the core-shell particle will have much greater absorption than the core in isolation. However, the carbon core is still responsible for all of the absorption in a core-shell particle, as the water is not absorbing any photons.

Thus, the conceptual difficulty of attributing absorption to different species in an internal mixture should not be that difficult - we can compute the effect of any absorber in a model by removing it. For example, when we compute direct aerosol radiative effects, we compute the difference between "with aerosols" and "without aerosols." Likewise, direct aerosol forcing is computed by differencing the present-day aerosols with an estimate of the aerosol loading in some reference year (e.g., 1850). Why not do the same thing with individual aerosol species? Thus, the AAOD effect caused by a target species is: (total aerosol absorption) - (aerosol absorption in the absence of the target species). The resulting AAOD represents the physical impact of omitting the target species from the atmosphere, which is what we are seeking. Glancing through Table 2, it seems that CAM5-ATRAS is the only model that got this right (good job CAM5-ATRAS!).

Some of the AeroCom modelers have lamented in the past that this method of speciating absorption alters the aerosol size distribution, but so what? Our traditional computations of DARE and DARF dramatically alter the aerosol size distribution, too. In the case of DARE, after all, it totally removes the size distribution! Why is it folly to remove part of a size distribution instead of all of it? The other definitions in the table, including the old Aerocom protocol* recommendation at https://aerocom.met.no/protocol expl.html, do not represent anything physical regarding internal mixtures and do not recognize that the absorption of an internal mixture is often greater than the sum of the absorbing components. In addition, the AeroCom protocol prescribes a set of densities that may or may not be consistent with any particular model. Thus, it seems as though this protocol was created mainly because of its simplicity.

*Aerocom protocol at https://aerocom.met.no/protocol expl.html:
Aerosol optical depth per species for aerosol internal mixtures: A check on how you compute the contribution of a given aerosol species to total aerosol optical depth if you have internal mixtures in the model: Procedure recommended during AEROCOM workshop: Compute volume fraction of aerosol species in aerosol particle volume (without water!!) and retrieve with that fraction the aerosol optical depth for a given species. Apply the following densities for the major species (Dust = 2650 kg/m3 / Sea salt = 1600 / Sulfate = 1769 / Black Carbon = 1500 / Particulate organic matter = 1500 )

Given the wide variety of methods for attributing AAOD to the absorbing species in this paper, the values for the component AAODs in Table 3 are not comparable between models. Since optical properties are computed offline, it would not be that difficult for each modeler to compute component AAODs with a single sensible protocol (ideally, as I described above for the internal mixtures) with a single set of refractive indices and densities for the absorbing aerosols. Constraining all of the models in the same way like this would make this paper strong and valuable. Otherwise, the paper just reports what we basically already know – the models are different from each other and the observations, and we have not quantified why.

Response: We appreciate this very thorough, constructive, and helpful review. We have done the following major changes to the manuscript:

1) Several of the models have applied the suggested method by removing one absorbing component by time completely in new model simulations. This method works in many of the models, but in the most sophisticated models it can provide erroneous results (e.g.

negative AAOD for some of the absorbing components): Therefore we have added the following text:

*For models with external mixing, it is straight forward to estimate specie-specific absorption. The mass absorption coefficient (MAC) for any specie is estimated using Mie theory and is a function of density, size distribution and the imaginary component of the complex refractive index at a given wavelength. For models with internal mixing, the estimated absorption per specie is more conceptually difficult because the sum of the absorption for each specie does not equal the sum of the internal mixture. For this study, the models with internal mixing, when possible, have used the same method for estimated specie-specific absorption; by removing the target specie and estimating the total absorption between the control run and the run with the specie removed. Even if this changes the size distribution of the aerosols for an internal mixture of an absorbing aerosol and a purely scattering aerosol this is an appropriate and accurate approach since the absorbing compound causing all the absorption. However, an internal mixing of absorbing aerosols causes changes in the size distribution of other absorbing aerosols and thus this method yields an inaccurate result for absorption of an individual aerosol. Therefore, for some models with sophisticated aerosol microphysical schemes the individual aerosol absorption is not reported.*

2)    The Reviewer points out that several models use outdated values for refractive index, size distribution, and density for BC compared to current knowledge. Additionally, he acknowledges that using recommended values in Mie calculations, which require assumption of spherical particles, give mass absorption coefficient MAC lower than in observations. Therefore, modelling using Mie theory requires assumption of input to Mie calculations outside recommended values to arrive at MAC values close to observation. The approach differs between models, but the BC MAC value is what has the clearly largest importance. In the response to the suggestion to unify the input to Mie calculation of refractive index, size distribution, and density we have added an important sentence:

*The actual choice of refractive indexes and density plays a minor role since it should be constrained by BC MAC recommended value of 7.5 $m^2$ $g^{-1}$. In models having a BC MAC for external mixed BC much lower than 7.5 $m^2$ $g^{-1}$ the aerosol optical properties should be updated based on current knowledge.*

3)    We have performed new Mie calculations for externally mixed BC in models where required information is available. The results show that models which include internal mixing enhance the MAC compared to the external mixing at emission. Several of the models with external mixing have a low MAC compared to measurements.

**Table 2**
I really like the idea of Table 2 because it is a great idea to have the mixing assumptions for the AeroCom models all in one place. I don't understand some of the methods and terminology, though. For instance, what does "*Core-shell for internally-mixed BC particles; Volume mixing for pure BC and BC free particles*" mean for the CAM5-ATRAS model? A reader might surmise from this text that the authors are using volume-averaged refractive indices for external mixtures, which does not make sense.

Response: For internally-mixed BC, BC makes the core and non-BC species make the shell (shell is assumed to be mixed well). For pure BC, BC refractive index is used for optical

calculations. For BC free (non-BC) particles, all non-BC species are assumed to be mixed well, using volume-averaged refractive index.

"Well-mixed for BC free particles" may be better than "Volume mixing for pure BC and BC free particles". We have updated Table 2.

It is also not clear which components are being internally mixed in many of the models. For example, ECHAM-HAM, ECHAM-SALSA, and GISS-MATRIX do not indicate which absorbing species they are treating as internal mixtures (BC, OA, dust, or some combination of all 3).

Response: All species in ECHAM-HAM, ECHAM-SALSA and GISS-MATRIX can be internally mixed. We have clarified this in Table 2.

I was surprised to see that the EC-Earth3 model treats sulfate, ammonium-nitrate, organic aerosols, sea salt, and water as homogenous mixtures described by the Bruggeman mixing rule. This rule is usually applied to dry insoluble components, like mineral mixtures. The table seems to indicate that all of these components are mixed together into one internal mixture, but is that accurate?

Response: We thank the reviewer for raising this question. The refractive index of internally mixed particles in EC-Earth3-AerChem is indeed described as indicated in the Table. The mixing rule assumptions are based on the work of aan de Brugh et al. (2011); who introduced the Bruggeman mixing rule to calculate the refractive index also for internally mixed particles in the soluble modes. An alternative for the soluble modes would be to use a simple volume weighting of the refractive indices of the mentioned components. The developers of the model will review the pros and cons of both approaches for future versions of the model. Note that the relative impact of the assumed mixing rule is smaller for the real than the imaginary part of the refractive index and that what matters most for absorption is the treatment of black carbon and dust (e.g. Lesins et al., 2002), for which the Maxwell-Garnett mixing rule is used. Moreover, the assumed mixing rule won't affect the effective properties of a mixture consisting of sulphate and ammonium-nitrate mixture, which in the model are assumed to have the same refractive index.

Aan de Brugh, J. M. J., Schaap, M., Vignati, E., Dentener, F., Kahnert, M., Sofiev, M., Huijnen, V., and Krol, M. C.: The European aerosol budget in 2006, Atmos. Chem. Phys., 11, 1117–1139, https://doi.org/10.5194/acp-11-1117-2011, 2011.

Lesins, G., Chylek, P., and Lohmann, U., A study of internal and external mixing scenarios and its effect on aerosol optical properties and direct radiative forcing, J. Geophys. Res., 107( D10), doi:10.1029/2001JD000973, 2002.

The SPRINTARS model describes internal mixing of BC with OC in the 2nd column of Table 2, but the third column says that the BC AAOD is calculated assuming that all BC is externally mixed. Thus, this is inconsistent.

Response: In SPRINTARS, the internal mixture of OC and BC is applied to the transport and radiation processes except that the 50% mass of BC originating from fossil fuel consumption is externally mixed. In SPRINTARS the BC AOD and AAOD cannot be calculated with assumption of internal-mixed particles of BC/OC. They are calculated assuming external mixing for convenience.

The NorESM2 model states:
Internal and external mixing. Maxwell-Garnett is used for calculation of refractive index of internal mixing of BC with other components, otherwise volume mixing. What is volume mixing? If the modeling is applying volume averaging of refractive indices to external mixtures, then this is not the correct way to compute optical properties of external mixtures. Thus, "volume mixing" for external mixtures needs to be defined.

Response: "Otherwise volume mixing" here refers to internal mixtures of non-BC aerosols; sulfate, sea-salt, organic matter, and dust. Finally, if BC is present the Maxwell-Garnett rule is used for BC vs. the rest (which consists of sulfate etc., i.e. the less absorptive components). We have updated Table 2.

**Figures**
Figure 1 is based upon Mie calculations taken from Samset et al. (2018). However, Samset et al. (2018) provides information about the size distributions, refractive indices, and densities that are required to make this figure, whereas this article does not. Additionally, this article fails to explain that the Bond and Bergstrom (2006) recommended value of MAC = 7.5 m2/g can not be achieved using the Bond and Bergstrom (2006) recommended refractive indices and densities, which explains why the "fresh" symbols lie outside of the shaded region. Similarly, the "coated" and "collapsed, uncoated" symbols are left unexplained in this figure. Samset et al. (2018) explains that the symbols are based upon values found in the literature and lists several citations.

These are important details because a reader should be able to understand all figures on the basis of the material in the article. Additionally, since the caption states that MAC is separable amongst the species in Figure 1, the reader needs to know the range of size distributions and densities associated with the shaded regions.

Response: Thanks for the comment. The calculations behind this figure rest on a number of assumptions, as the reviewer points out, and we have to refer to the original publications for the details. However, we agree that more description should be given than what was in the original manuscript. The following has been added to the discussion of Figure 1, in the Introduction and figure caption:

*Figure 1 illustrates how the dependence of the mass absorption coefficient (MAC) on wavelength differs between these three major species of absorbing aerosols (Samset et al. 2018). It shows both observations (shaded bands) and Mie calculations made with parameters from recent literature. Briefly, size distributions for BC and BrC had a radius and sigma of 0.04 µm and 1.5 for BC, and 0.05 µm and 2.0 for BrC, while for mineral dust, they used observed sizes from the DABEX aerosol campaign. Aerosol densities were 1.2, 1.8, and 2.6 g cm−3, for BrC, BC, and dust, respectively. For BC, the figure also shows additional MAC values (gray circles) where the Mie calculations have been scaled to achieve the value of 7.5 m2 g−1 at 550 nm recommended in Bond and Bergstrom 2006, as well as range of values found in the literature for coated BC, and collapsed, uncoated BC. For further details, see Samset et al. 2018.*

**Figure 2**: Given that this paper is about modeling aerosol absorption, some discussion about the use of OPAC optical properties is warranted. Six of the 14 models in Figure 2 use BC imaginary refractive index (BC IRI) of 0.44, which is the value given by OPAC. That means that nearly 1/2 of the models are still using OPAC, but more than 1/2 of the models have

figured out a way to move beyond OPAC. Way back in 2006, Bond and Bergstrom (2006) reported this in their assessment of BC refractive indices: "The value commonly used by climate modelers (m = 1:74 □ 0:44i at 550 nm) represents none of the possible refractive indices and should be retired."

Bond and Bergstrom (2006) also said:
"The history of refractive index values tabulated by Shettle and Fenn (1976, 1979) is worth special mention. These are by far the most prevalent values for use in climate modeling, and have been incorporated into widely-cited literature, including a book by d'Almeida et al. (1991), and the Optical Properties of Aerosols and Clouds (OPAC) program (Hess et al. 1998). The original work by Shettle and Fenn (1976) averaged values from an earlier review by Twitty and Weinman (1971). In turn, the averaged data are taken from McCartney et al. (1965), who measured three coals, and Senftleben and Benedict (1918), who reported soot generated from an arc lamp. The review does not incorporate most of the findings on soot in the combustion literature, and indeed was written before most of that work was available. The precision of both n and k provided in OPAC values (three decimal places) is unwarranted, given this history. The OPAC value of 1.74-0.44i is drawn from incompletely graphitized carbon and has a lower value of k than most soot. . .
. . . The optical and physical data for LAC propagated by d'Almeida et al. (1991) have some interesting properties. Along with an imaginary refractive index that is too low, these authors recommend: (1) a particle size that is far too small (23 nm is the approximate size of primary spherules, not aggregates); (2) a geometric standard deviation that is somewhat too large (2.0); and (3) a density that is far too low (a density of 1.0 is never observed; Fuller et al. (1999) tabulate measurements indicating densities of about 1.8 g/cm3). Despite returning to the string of citations that led to d'Almeida et al. (1991), we have been unable to unearth the sources of these values. When compared with measured values, each of the individual assumptions above may lead to an error of 50-75% in calculated properties that affect climate forcing."

It is unfortunate that so many models did not follow the Bond and Bergstrom (2006) recommendations. Since the present article is about model diversity of absorption, this would clearly be a great place to revive the OPAC BC issue.

Response: We have removed figure 2 and instead made figures that compare AAOD, MAC, load, density, and refractive indicies, as suggested. We have included a discussion around the values of BC refractive index in Fig 4, highlighting the recommendation from Bond and Bergstrøm (2006). See main comment. In Myhre et al., (2009) the values in the OsloCTM model were compared with values in Bond&Bergstrom. The density numbers had to be tuned to achieve a MAC around 7.5 m2/g. The radiative forcing estimates were almost identical.

Myhre, G., Berglen, T. F., Johnsrud, M., Hoyle, C. R., Berntsen, T. K., Christopher, S. A., Fahey, D. W., Isaksen, I. S. A., Jones, T. A., Kahn, R. A., Loeb, N., Quinn, P., Remer, L., Schwarz, J. P., and Yttri, K. E.: Modelled radiative forcing of the direct aerosol effect with multi-observation evaluation, Atmos. Chem. Phys., 9, 1365–1392, https://doi.org/10.5194/acp-9-1365-2009, 2009.

**Comparisons to AERONET data**
Given the text on lines 192-200, does it even make sense to compare any extrinsic AERONET properties (like AAOD) to the modeled values, as in Figure 4?? That is, the AERONET dataset that the authors are using require AOD(440) > 0:4 (per line 193), but the models use all available AOD (per line 195). So why would there be an expectation that any modeled

extrinsic parameter would compare well with AERONET under these conditions (even with a high-resolution grid)? If the authors checked the coincident-only AODs that are obtained during the same scans that produce the AAODs, I expect that they would see a significant bias in AOD for the same reason - modeled extrinsic parameters will not compare well to AERONET when you discard low-load cases in AERONET but keep low-load cases in the model output. (If they don't see a bias, that means that they are not using enough AERONET sites with AOD(440) < 0:4, or something is very wrong with the model). In fact, model comparisons to AERONET should be required to show coincident-AOD comparisons whenever there is a comparison of extrinsic parameters (like AAOD).

I suggest the following. . . plot aerosol co-albedo (absorption/extinction) instead of AAOD. Co-albedo is a pseudo-intrinsic parameter that is much less susceptible to column aerosol loading than AAOD.

Response: we have removed the comparison of AERONET.

**Unclear Definitions for Organic Aerosol, Organic Carbon, and Brown Carbon**
The authors are not clear about the differences between OA (organic aerosol), organic carbon, and brown carbon. Indeed, the text doesn't seem to recognize that BrC is a subset of OC and that OC is a subset of OA. On line 66, the authors refer to BrC/OA, implying that BrC and OA are one and the same species. On line 117 and in Table 2, though, the authors discuss OA/OC ratios, which are always greater than 1. On line 124, the authors mention that the *"OsloCTM3 model divide OA into into a mix of absorbing and non-absorbing species. . . "*.

Response: we have added text in the introduction to better clarify the differences between BrC, OA and OC.

The authors need to inform their readers that the brown shaded regions in Figure 1 correspond to organic aerosols that are "washed" with solvents like acetone and methanol in order to extract the absorbing organic aerosols from the non-absorbing organic aerosols. The OsloCTM3 model seems to recognize this by separating absorbing and non-absorbing OC. The other models apparently use a single set of optical properties that encompasses all organics (both absorbing and non-absorbing), so they should be using MACs that are much lower than the values shown in Figure 1. Additionally, BrC is produced by hulis and is largely found in biomass burning and is not observed in fossil fuel burning. Thus, the models should be using a different set of optical properties for OC that originate from modern urban areas than the optical properties that they use for biomass burning. None of these concepts are clearly recognized or discussed in the text.

Response: we have added more text in relation to the new Figure 6 and in the Introduction (the changes in the text are too substantial to be copied in here, so I hope it is OK that we refer to the main manuscript for the actual changes, - this related to this comment and several others).

**Wavelength Dependence of Absorption**
Section 3.4 and Figure 11 are another missed opportunity for some good discussion. All of the models except for OsloCTM3 show significant OA absorption at 870 nm, but Figure 1 indicates no detectable BrC absorption at this wavelength. Clearly, none of the models are using the recent measurements for BrC that are shown in Figure 1 (except perhaps OsloCTM3 - good job Oslo!). This begs the question - what are the models using for OA? Given that this is a paper on understanding the model spread of AAOD, this is an important topic. Sure, the

authors provide OA refractive indices at 550 nm in Figure 2, but what measurements are the basis of the refractive indices in the models, and why are the modeled OA MACs and dust MACs buried in the supplement? The OA MACs in Fig S8 are on the low end of Fig 5 (assuming that the wavelength in S8 is 550 nm, but that is not stated), so does that mean that the OA imaginary refractive indices are more or less constant wrt wavelength? Also, the OA MAC varies by a factor of 4+ in Figure S8 - how come this huge range of modeled optical properties for OA is not discussed in the text? Same question for the MAC of dust, which varies by a factor of 3+ in Figure S8.

Response: we have moved OA and dust MAC into the manuscript and added a discussion of the same parameters as for BC (new Fig 6 and Fig 8). The spread between the models is less than previous using the new method where NorESM now is not an outlier. See main comment above.

**Going Forward**

There is potential for an excellent paper in this material. I would like to see:

- An analysis of the models with external mixing. What does the spread in MACs and AAODs for this subset of models look like, and what is the cause of the spread? Size, refractive index, and density are the important parameters. Analyzing MAC wrt IRI/density will probably be enlightening. Column mass loading of the absorbing aerosols should be a factor.

  Response: We have made new figures (bar graphs) for each species, with AAOD, MAC, load, density, and refractive index. When possible, we have estimated MAC using Mie theory (size distr, refractive index and density) for externally mixed BC, this is included in the last column. The models with external mixing are marked with a grey background. We have added new text in the manuscript with analysis for the two model subsets.

  We have looked at correlations between parameters, but it makes less sense to include this in the manuscript: For the models with BC internal mixture, the correlations between BC AAOD and load are high (0.9), while the correlation between BC AAOD and density and between BC AAOD and refractive index is low (0.1-0.2). On the other hand, for the models with external mixing, the correlation between BC AAOD and density/refractive index is high (-0.7/0.8).

  For OA AAOD in the four models with internally mixed OA, the correlation with load is 0.6, and the correlation with density is -0.8. For OA AAOD in the models with externally mixed OA, the correlation with load is 0.5 and with density is low (0.2).

  The correlation is high for dust AAOD in the models with external mixing with column load (0.97) and density (0.74), and no correlation for the refractive index (-0.03). The correlation is lower for the corresponding correlation in the models with internal mixing (0.3 for load, 0.4 for lifetime, 0.6 for density and 0.4 for refractive index).

- A single optics module applied to the models with internal mixing, as much as possible. Use the CAM5-ATRAS method for splitting absorption amongst components and analyze the spread in MACs and AAODs, and abandon the old AeroCom protocol.

Response: Actually, CAM5-ATRAS removed one and one species, but calculated this using offline optical calculations in a simulation. CAM5-ATRAS did not make online simulations with changing emissions (removing emissions from an absorbing species). This is a sophisticated way to estimate absorption, but not many models have these calculations available in their radiation code. The models with internally mixed BC have made new simulations using the suggested method. → i.e. NorESM, ECHAM-HAM, ECHAM-SALSA, and GFDL. We have updated Table 2. However, for ECHAM-SALSA, using this method resulted in a negative AAOD for OC; removing OC reduces the size of BC. When OC is removed, the volume of BC will not change, but it's size will change since it is internally mixed with OC. The volume absorption cross section will increase, and the same amount of BC becomes more absorptive. We therefore kept the 'old' method for ECHAM-SALSA and added a discussion around this.

- Pay attention to the mass/volume fraction of absorbers in the mixtures, especially for BC. Here again, column mass loading will be important for understanding AAOD.

  Response: we have included BC column mass loading as one of the panels in new Fig 4 for BC, new Fig 6 for OA, and new Fig 8 for dust.

- With separate analyses of the external and internal mixtures now in hand, how do these two analyses compare to each other? How do they compare to the models with internal mixing that are not amenable to the approach in the 2nd bullet?

  Response: we have separated the models with internal and external mixing in Fig4, 6 and 8 and added a discussion around this.

- Don't compare to any extrinsic AERONET parameters, since the AOT(440) > 0:4 restriction skews the AERONET data to large aerosol loadings. Use AERONET's single-scatter co-albedo, or omit.

  Response: we have removed the comparison with total AAOD and AAE from AERONET.

- Pay attention to BC density. The use of OPAC BC with a realistic density (as opposed to OPAC's recommended value of 1 m2/g) is probably the cause of the lowest MACs in Figure 9. This could be easily remedied by abandoning OPAC. The densities of all absorbers should be included in a table.

  Response: We have included density in the bar chart for each species (new Fig 4, Fig 6, and Fig 8).

- Abandon OPAC. Since the authors are only considering a few wavelengths (440, 550, and 870 nm), abandoning OPAC is not a tall order.

  Response: Please see main comment. We have made a clear recommendation that models having a BC MAC value low compared to observations, should update their optical property scheme. Similarly, for models with a low OA AAE we have made a recommendation to use current knowledge based on a large set of observations. However, for BC and OA there is substantial uncertainty in refractive index and density, simply because of the large variability in their optical properties.

Inform the reader about where each model obtains their information for optical properties of aerosols (i.e., provide citations). Many of them use OPAC, but where do the other models find complex refractive index, size, and density information? This will allow the reader to understand which models are using the latest measurements, and which ones are lagging.

Response: We have pointed to references for OA and its strong wavelength dependence. We have already referred to Bond and Bergstrom for BC (see also main response). Furthermore, a description of details behind Fig 1 is now included in the supplementary

- A description of the size distributions that the models use is needed. Lognormal radii, widths, etc., should be tabulated.

This is a difficult task and hard to do consistently. For instance, here is some information for ECHAM-HAM-M7: The log-normal modes for M7 are described in Table 1 of Stier et al. (2005). Mode median radii are allowed to vary between the given ranges. This is often misunderstood: as the log-normal is unbound the distribution also includes radii out of the ranges given (and the Mie calculation is done based on the full distribution). Mode-merging redistributes between the modes. The actual size will of course vary spatiotemporally.

And here is for ECHAM-SALSA: uses sectional size classes starting from 3 nm in diameter to 10 um (although the largest size class has no upper limit and can grow as large as they want). There are 3 size classes between 3 nm - 50 nm and 7 between 50 nm - 10 um. Size classes between 50 nm - 10 um have two parallel externally mixed size classes, one for soluble compounds and insoluble compounds. Black Carbon and Dust are emitted to insoluble size classes but can get mixed with soluble particles through coagulation.

Perhaps the size distribution in the AeroCom models is a study on its own?

We therefore refer to the model's own papers for this information.

- Make sure that the figures are self-contained (in the sense that the reader does not have to go to another paper to understand how they were created).

Response: we have added more information about Fig 1 (and removed AERONET, if this comment also referred to those?)

- Clear up the phraseology everywhere, especially in Table 2 where it is so important.

Response: we have rephrased some of the expressions in Table 2.

- Make sure that all text is clear so that a reader understands the intent of your words without ambiguity. There are many co-authors on this paper for proofreading, so this should not be an issue.

- Clearly define OA, OC, and BrC early in the text.

Response: We have included a clear definition of the three in the Introduction.

All of this should be do-able with the model runs already utilized in this paper.

**Line-by-line issues that need to be addressed:**
Line 70:
*"While fine-dust particles mostly scatter solar radiation, coarse dust also absorbs moderately in the visible and near-infra-red spectrum.*" I would like to see a citation for this, because I am not sure if it is accurate. Iron oxides in dust are small, so I don't see why they would not be present in the fine-mode dust as well as coarse mode dust.

Response: we have added two references:
Ryder, C. L., Marenco, F., Brooke, J. K., Estelles, V., Cotton, R., Formenti, P., McQuaid, J. B., Price, H. C., Liu, D., Ausset, P., Rosenberg, P. D., Taylor, J. W., Choularton, T., Bower, K., Coe, H., Gallagher, M., Crosier, J., Lloyd, G., Highwood, E. J. and Murray, B. J. (2018) Coarse mode mineral dust size distributions, composition and optical properties from AER-D aircraft measurements over the Tropical Eastern Atlantic. Atmospheric Chemistry and Physics, 18. pp. 17225-17257. ISSN 1680-7316 doi: https://doi.org/10.5194/acp-18-17225-2018

Ryder, C. L., Highwood, E. J., Rosenberg, P. D., Trembath, J., Brooke, J. K., Bart, M., Dean, A., Crosier, J., Dorsey, J., Brindley, H., Banks, J., Marsham, J. H., McQuaid, J. B., Sodemann, H. and Washington, R. (2013) Optical properties of Saharan dust aerosol and contribution from the coarse mode as measured during the Fennec 2011 aircraft campaign. Atmospheric Chemistry and Physics, 13 (1). pp. 303-325. ISSN 1680-7316 doi: https://doi.org/10.5194/acp-13-303-2013

Line 157:
*". . . their imaginary parts of the refractive index vary a lot (1.75 + 0.44i for SPRINTARS and 1.85 + 0.71i for GISS-MATRIX (Fig. 2).*" This is a missed opportunity - why not use the same BC refractive index and BC density in all of the models? This would eliminate a significant source of diversity so that part of the analysis could be focused on the remaining causes of diversity (i.e., size distribution and column mass loading). Additionally, the diversity of BC density is entirely missing from this paper. BC density is very important because it is inversely proportional to MAC (all else being equal), and therefore has a direct impact on both MAC and the AAOD computed by the models. Finally, why not compare column mass loading amongst the models as well? I believe that we could learn something by studying mass diversity alongside the MAC, AOD, and AAOD diversity, but column mass loading was not even mentioned as a source of diversity in this article.

Response: We are a bit confused about this comment. Does the reviewer suggest that all models should change the radiation code for this study? It is easier said than done to make all models change their radiation code. Also, many of the models are tuned to get a satisfying MAC.

We have included density and mass load as part of the discussion.

Line 199:
*"However, using a high-resolution simulation of global aerosols, Schutgens (2020) found a much smaller bias of 9%"* Ending the paragraph of caveats with this sentence is a bit of a hoodwink, as the present study does not use a high-resolution model.

Response: this sentence has been removed from the manuscript as part of the new discussion added.

Line 213:
"*In NorESM2 the additional absorption is from sea-salt and sulfate (mixed with BC, dust and OA). In GFDL BC is internally mixed with SO4, so the additional absorption is due to SO4 (mixed with BC). . .* "

This is not explained very well because the most important part of the explanation is in parentheses. Reading this, one might think that sea-salt, sulfate, etc. are absorbing photons. The authors should clearly point out that internally mixed BC has greater absorption than externally mixed BC, and that this is because the internal mixtures have greater geometrical cross-sectional areas than the BC inclusions within the mixtures. The sea-salt and sulfate in the mixtures are not absorbing any photons, whether they are hosting absorbing inclusions or not. This needs to be clear.

Response: We have removed this sentence, because both GFDL and NorESM have made new simulations using the same method as CAM5-ATRAS. We have also pointed out the fact that internally mixed BC has higher absorption in the Introduction, Methods and when discussing Fig 4. We give numbers for AAOD BC averaged separately for internal and external mixed models (0.0043 vs. 0.0015, respectively).

Line 323:
The authors mention BC particle size as 50 nm, but they do not state whether this is radius or diameter. Additionally, 50 nm is closer to the diameter of a spherule than a BC aggregate. For example, Schwarz et al. (2008) measured size distributions of BC with an SP2, and they obtained a median diameter of fi200 nm.

Response: to be more accurate we have rewritten the sentence to: '*Since most BC particles are in the fine mode with wavelength-independent index of refraction over the visible spectrum*'

Line 363:
If the global mean modeled AOD is 0.129 and the global model mean AAOD = 0.035 (per line 188), then the global mean SSA should be close to $(1 - .035/.129) = 0.73$. Why is the global mean AAOD so high wrt to the global mean AOD?

Response: There was a zero missing. And SSA = Scattering AOD / AOD = $(0.129 – 0.0035) / 0.129 = 0.97$. The comparison with AERONET has been removed. The total AAOD is 0.0054.

Line 391-393:
"*We also find very little correlation between the imaginary index and mass absorption coefficients. For BC just three different refractive indices are used by the models, while the spread is not related to this choice.*"

This sentence appears in the Summary, but I don't see IRI/MAC correlations discussed in the paper. However, it is not surprising that the authors do not see a relationship between IRI and MAC if they did not account for the different densities used in the models. For instance, OPAC uses a density of 1 g/cm3, which results in a MAC of ~10m2/g (per Bond and Bergstrom, 2006). A more realistic choice of 1.8 g/cm3 for BC density would reduce the MAC to $10/1.8 = 5.5$ m2/g. The imaginary refractive index makes a huge difference to the BC MAC, but not if one simultaneously tunes the BC density.

Response: this is exactly why it is difficult to draw firm conclusions based on the analysis of density and refractive indices, because models tune these parameters. We have rewritten this sentence and added a discussion on BC refractive index in Fig 4.

Line 398:
*"We recommend that the role of size and mixing rules and diagnostic procedures should be investigated in more detail to understand the differences in mass absorption coeficients."* I agree completely, but why didn't the authors do that in this paper? That is actually what many readers will be looking for in this article.

Response: we have included more discussion about this in the Results chapter.

Line 403:
*"In particular, we have found that the imaginary indices are not explaining much of the AAOD variance, except slightly for dust."*
Here again, I did not see this discussed at all in the paper. The final section should be a summary of the details that are presented in the paper, not the introduction of a new result.

Response: we have included more discussion about this in the Results chapter.

**Minor issues:**
Line 51:
*"The three absorbing species are rarely observed as single species,. . . "*
This sentence does not make sense to me. . . Shouldn't this go unstated by definition?

Response:  We have added a reference to the sentence. It is there to illustrate that the models are not able to mix the aerosols as observed in nature.
*The three absorbing species are rarely observed as single species (Fierce et al., 2016), while many models are not able to fully mix the aerosols and therefore treat them as separate species in an idealized way with their own life cycles and optical properties*

Line 58:
*"but these calculations are approximate (using mixing rules or the assumptions of a co-centric core/shell structure). . . "*
This sentence seems to imply that internal mixing requires additional significant assumptions wrt external mixing, which is not the case. Core/shell computations are exact, although the core/shell structure is an approximation for the shape of aerosols in the atmosphere. Likewise, Mie Theory is also exact for the spherical particle approximation that is used for all externally-mixed aerosols in the models (with the exception that some models use a spheroidal approximations for dust), but spheres and spheroids are still approximations for particle shape. Thus, approximations associated with shape are required for both internal and external mixing. Finally, errors associated with efective medium approximations have been tested by many authors (e.g., Martins et al., 1998; Fuller et al., 1999; Lesins et al., 2002) and are likely swamped by other modeling errors associated with mixing fractions and assumed component refractive indices.

Response: these calculations will be approximate because of the uncertainty in the mixtures containing different absorbing aerosols and/or water, or if the BC core is mixed with mineral dust for instance.

Line 61: Begins with "*However,...*" and then essentially repeats the information on line 58. Sentence should start with a different word.

Response: we have removed the word.

Line 102: *"the separation into fine (< 1μm) and coarse mode (> 1μm) AOD..."* Presumably the authors mean diameter, but this should be specified.

Response: We have added this to the sentence.

Line 119: *"and 11 models have provided absorption split into BC and dust (OA)."* Why is OA in parentheses?

Response: This was a typo; we have rewritten the sentence.

Line 147: Authors are using Version 2 AERONET, but which level? That is, Level 1.5, or 2.0?

Response: we have removed the comparison with AERONET.

Line 183 (Fig 4) and Line 185: The authors need to specify whether they are using Version 2 or Version 3 AERONET and Level 1.5 or Level 2, here. The authors mention Level 2 later on Line 193, but it is not clear whether they realize that Version 2 means something different than Level 2.

Response: we have removed the comparison with AERONET.

Line 228: Table 2 says that Sprintars has internal mixtures of BC with OA. Why do the authors include it as a model with external mixing?

Response: SPRINTARS treats the absorption as externally mixed BC and therefore we have labelled it 'externally mixed'.

Line 276: Replace "Methods" with "Section 2 - Methods" for improved reader navigation.

Response: this has been added.

Line 325 & 328: BC AAE was addressed in Schuster et al. (2016b), not Schuster et al. (2016a).

Response: we have added this reference.

Line 335: Should be Fig 13, not Fig 10.

Response: We have changed this.

Line 360:
"Compared to retrieved AAOD from AERONET stations, the models yield lower absorption." One would hope so, since the AERONET dataset that the authors use requires AOD(440) > 0.4, whereas the models are not using the same restriction.

Response: we have removed this comparison.

Line 381: It is good that that the authors are recommending and update for modeled OA, but it would be nice if they reported what the models are actually using right now, too.

Response: we have extended the discussion of refractive indices for OA in new Fig 6.

References

Bond, T., and R. Bergstrom (2006), Light absorption by carbonaceous particles: An investigative review, Aerosol Sci. Technol., 40 (1), 27{67.

Bond, T., et al. (2013), Bounding the role of black carbon in the climate system: A scientific assessment, J. Geophys. Res., 118, 1{173, doi:10.1002/jgrd.50171.

Fuller, K., W. Malm, and S. Kreidenweis (1999), Efects of mixing on extinction by carbonaceous particles, J. Geophys. Res., 104 (D13), 15,941{15,954.

Koch, D., et al. (2009), Evaluation of black carbon estimations in global aerosol models, Atmos. Chem. Phys., 9, 9001{9026.

Lesins, G., P. Chylek, and U. Lohmann (2002), A study of internal and external mixing scenarios and its efect on aerosol optical properties and direct radiative forcing, J. Geophys. Res., 107 (D10), 4094, doi:10.1029/2001JD000973.

Martins, J., P. Artaxo, C. Liousse, J. Reid, P. Hobbs, and Y. Kaufman (1998), Efects of black carbon content, particle size, and mixing on light absorption by aerosols from biomass burning in Brazil, J. Geophys. Res., 103 (D4), 32,041{32,050.

Samset, B., C. Stjern, E. Andrews, R. Kahn, G. Myhre, M. Schulz, and G. Schuster (2018), Aerosol Absorption: Progress Towards Global and Regional Constraints, Curr Clim Change Rep, 4 (65), doi:10.1007/s40641-018-0091-4.

Sato, M., J. Hansen, D. Koch, A. Lacis, R. Ruedy, O. Dubovik, B. Holben, M. Chin, and T. Novakov (2003), Global atmospheric black carbon inferred from AERONET, Proc. Natl. Acad. Sci., 100 (11), 6319{6324.

Schuster, G., O. Dubovik, and A. Arola (2016a), Remote sensing of soot carbon { Part 1: Distinguishing difierent absorbing aerosol species, Atmos. Chem. Phys., 16, 1565{1585, doi:10.5194/acp-16-1565-2016.

Schuster, G., O. Dubovik, A. Arola, T. Eck, and B. Holben (2016b), Remote sensing of soot carbon { Part 2: Understanding the absorption fiAngstrom exponent, Atmos. Chem. Phys., 16, 1587{1602, doi:10.5194/acp-16-1587-2016.

Schwarz, J., et al. (2008), Measurement of the mixing state, mass, and optical size of individual black carbon particles in urban and biomass burning emissions, Geophys. Res. Lett., 35, L13810, doi:10.1029/2008GL033968.

---

## Author Comment (AC2)

**Referee report regarding the manuscript: Aerosol absorption in global models from AeroCom Phase III**

Authors: M. Sand et al.

General comments

In my opinion this manuscript is not suitable for publication in Atmospheric Chemistry and Physics. I do not think it contains enough new and interesting material within the Aims & Scope of ACP. The area of aerosol absorption is certainly of interest for ACP, but the manuscript is mainly a model intercomparison of technical character, with no clear scientific conclusions or substantial new concepts, ideas, methods etc. regarding the subject of the paper.

It would be more suitable as a Technical report (or it would have been useful as Supplementary material to the large AeroCom III Model intercomparison already published in ACP; Gliß et al., 2021, https://acp.copernicus.org/articles/21/87/2021/acp-21-87-2021.html).

Since some of the material presented in the manuscript could be of some interest to other modellers it could perhaps have been acceptable in a more technical journal (possibly Geoscientific Model Development), if a much more substantial discussion of the results was added, but since it has already been published as a preprint in ACPD — and will thus remain permanently archived, citable, and publicly accessible — I do not think a submission to GMD would be worthwhile. The Preprint in ACPD can, in a sense, be considered an "extra Supplement" to the article by Gliß et al. (2021).

My suggestion is thus that the manuscript is not accepted for publication in ACP.

*Response: We thank the reviewer for reading the manuscript. We disagree with the reviewer that this should be a supplement to Gliß et al. (2021). The Gliß et al. (2021) paper concerns a general description of optical properties of the AeroCom models, while in this study we document the absorption in detail, and we look at different wavelengths and for different species. It is important to document this in a separate paper, and not as a supplement to an already large, published paper. We have rewritten the manuscript based on recommendations from the other reviewer, focusing on causes of the spread in absorption among the models. The manuscript in its current form is the result of a lot of work, analysis, and discussion among the modelers. We get the impression that the reviewer simply does not seem to see the point of model intercomparisons and feel that the feedback is less than constructive. We hope that the substantial changes we have done to the paper based on the other reviewer's comments, will make this manuscript within the scope of ACP.*

Since I do not think the manuscript is suitable for ACP I have not made a full in-depth review of all the details, but I noted a few minor things when reading it, and I list these below.

Some specific comments

Page 2, line 60: Stier et al. 2017 — I guess this should be Stier et al. 2007

*Response: Yes, thanks, this has been changed.*

Page 3, line 69: Textor et al., 2006 — should be Textor et al., 2007?

*Response: Yes, thanks, this has been changed.*

Page 4, line 97: Randells et al. 2013 — missing in reference list?

*Response: we have removed the citation.*

Page 5, line 119: What do you mean by "dust (OA)"?

*Response: there should also have been a number 10 in bracelets, explaining that 10 models reported OA, while 11 models reported dust and BC. This has been changed.*

Page 6, Table 1 is essentially a copy of Table 2 in Gliß et al. (2021) — not necessary to duplicate here (a reference to Gliß et al. would be enough)

*Response: we think it is convenient to have the table in both papers, and because there are minor differences from Gliß et al. (2021). For instance GISS-MATRIX is not in Gliß et al. (2021). .*

Page 6, Table 1: References Bauer et al., 2008 and Bauer et al., 2020 are missing in the reference list

*Response: the two references has been added to the reference list.*

Page 10, Figure 3: The resolution of the subfigures is quite poor (at least on my screen)

*Response: we have made new figures.*

Page 10–11, Table 3: Half of the data in this table (BC MAC, BC Burden, BC lifetime, OA lifetime, Dust lifetime) were included already in Gliß et al. (2021) [Table 3]. Perhaps it is not necessary to include the same information here. However, some of the data clearly disagree with Gliß et al. (2021) and this needs an explanation:

*Response: We have removed Table 3 and made new figures focusing on burden, load, and refractive indices. We can refer to lifetime, if the same, in Gliß et al. (2021). The small differences are because there are slightly different model versions and some modelers have made additional runs, + for AAOD; we use all-sky only, while Jonas also use clear-sky for comparisons with observations. Jonas Gliß is a co-author of the study and we have closely compared numbers to avoid any errors.*

*We think it is better for the analysis to see the numbers for mass load (now included as bars in Fig 4,6 and 8), to better follow the discussion.*

Large difference for BC MAC for NorESM2; here 5.2 m2 g-1, but 3.2 m2 g-1 in Gliß et al.

*Response: this was because we added the coating effect from BC, as we explained in the paper (we did report both numbers in the manuscript).*

BC MAC for OsloCTM3 is 12.4 m2 g-1 here, but 13.0 m2 g-1 in Gliß et al.

*Response: this was because OsloCTM3 did an additional run after a minor update.*

BC lifetime in EMEP is substantially different here (2.2 days) compared to in Table 3 of Gliß et al. (2.9 days)

*Response: An additional run with EMEP as well.*

BC lifetime in TM5 8.6 days, compared to 8.4 days in Gliß et al.

*Response: see comment above*

OA lifetime in GFDL 4.1 days, compared to 4.5 days in Gliß et al.

*Response: see comment above - GFDL has also made new runs.*

Page 20, line 264 — the Section header is "BC MAC values" but this section also includes MAC for OA and dust.

*Response: this is no longer its own chapter as we have rewritten the manuscript.*

Page 20, line 275: Reference (Ytrri et al 2014) is missing in the reference list (and I suspect it should rather be Yttri et al?)

*Response: yes, this reference was only in the supplement, and has been added to the reference list with the correct name (Yttri).*

Page 20, line 275: Fig 8 — should be Fig 9

*Response: Yes, and the sentence has been removed (new figure).*

Pages 21–22 I found the discussion of "partial sensitivities" of AAOD to "variations in emission, lifetime, and MAC" confusing, and I do not see how it actually give any clear explanations of the AAOD differences between the models. A much more detailed discussion would be needed to understand this (I think). Also, I do not understand why Figure 10 is made as line plots? I think it just messes up the diagrams and make them less clear — perhaps bar diagrams would have been better?

*Response: We have made 3 new figures with bar diagrams illustrating MAC, load, density, and refractive index to better explain the model differences. We have also changed to text describing the figure to better describe the figure.*

Page 25, line 335: Fig. 10 — should be Fig. 13?

*Response: yes, and this have been changed.*

Page 25, lines 337–338: "Many of the AeroCom models have not updated their OA refractive indices to include BrC." — Be more specific! Which models have, and which have not, included the BrC?

*Response: we have added more discussion on this related to the new Fig 6. We have also rewritten the sentence: 'Most AeroCom models (except OsloCTM3 and GISS-OMA) have not updated their OA refractive indices to include BrC. BrC is mostly responsible for the spectral dependence.'*

Page 27, line 356: "and for the first time 11 (10) these models have reported" — What do you mean by (10)? Also, I guess there should be an "of" in the sentence (i.e. 11 of these models).

*Response: we have added this. We have removed '10'.*

Page 27, line 374: 8.6 m2 g-1 a [3.1-15.0] — remove a

*Response: this has been removed.*

Page 27, line 385: "BC lifetime (ranging from 4 to 9 days)" — I guess the range should be 2 to 9 days? According to Table 3 the BC lifetime in the EMEP model is only 2.2 days (but 2.9 days according to Table 3 in Gliß et al. 2021)

*Response: this has been changed.*

References (pages 29–36): In addition to the missing references mentioned above, a couple of references are not in the correct alphabetical order.

Supplement:

Figures S1, S3, S5 and S6 need to be larger or at least to be of better resolution.

*Response: we have removed the figures from the Supplement as they were redundant.*

Figure S2: The figure caption and legends lack information about the "fat" line and dots (measurements I assume).

*Response: we have removed the figure from the manuscript, but we kept the references in the supplement.*